# Movement-related coupling of human subthalamic nucleus spikes to cortical gamma

Petra Fischer[1,2]*, Witold J Lipski[3], Wolf-Julian Neumann[4], Robert S Turner[5,6], Pascal Fries[7,8], Peter Brown[1,2], R Mark Richardson[9,10]*

[1]Medical Research Council Brain Network Dynamics Unit, University of Oxford, Oxford, United Kingdom; [2]Nuffield Department of Clinical Neurosciences, John Radcliffe Hospital, University of Oxford, Oxford, United Kingdom; [3]Neurological Surgery, University of Pittsburgh School of Medicine, Pittsburgh, United States; [4]Department of Neurology, Campus Mitte, Charite - Universitaetsmedizin Berlin, Berlin, Germany; [5]Department of Neurobiology, University of Pittsburgh, Pittsburgh, United States; [6]Center for the Neural Basis of Cognition, University of Pittsburgh, Pittsburgh, United States; [7]Ernst Strüngmann Institute (ESI) for Neuroscience in Cooperation with Max Planck Society, Frankfurt, Germany; [8]Donders Institute for Brain, Cognition and Behaviour, Nijmegen, Netherlands; [9]Department of Neurosurgery, Massachusetts General Hospital, Boston, United States; [10]Harvard Medical School, Boston, United States

*For correspondence:
petra.fischer@ndcn.ox.ac.uk (PF);
Mark.Richardson@mgh.harvard.edu (RMR)

Competing interests: The authors declare that no competing interests exist.

**Abstract** Cortico-basal ganglia interactions continuously shape the way we move. Ideas about how this circuit works are based largely on models those consider only firing rate as the mechanism of information transfer. A distinct feature of neural activity accompanying movement, however, is increased motor cortical and basal ganglia gamma synchrony. To investigate the relationship between neuronal firing in the basal ganglia and cortical gamma activity during movement, we analysed human ECoG and subthalamic nucleus (STN) unit activity during hand gripping. We found that fast reaction times were preceded by enhanced STN spike-to-cortical gamma phase coupling, indicating a role in motor preparation. Importantly, increased gamma phase coupling occurred independent of changes in mean STN firing rates, and the relative timing of STN spikes was offset by half a gamma cycle for ipsilateral vs. contralateral movements, indicating that relative spike timing is as relevant as firing rate for understanding cortico-basal ganglia information transfer.

## Introduction

Oscillations in extracellular field potential recordings reflect the synchronous rhythmic activation of thousands of nearby neurons capturing a mixture of sub- and suprathreshold activity (*Buzsáki et al., 2012*). At the onset of contralateral movements, relatively narrow gamma-band oscillations (~60–90 Hz) appear in the human basal ganglia, thalamus and motor cortex (*Brücke et al., 2008*; *Brücke et al., 2012*; *Brücke et al., 2013*; *Muthukumaraswamy, 2010*; *Jenkinson et al., 2013*). The potential significance of these oscillations is illustrated by the fact that neural activity synchronizes strongly only when movements are actively initiated, but not during similar passive displacements (*Liu et al., 2008*; *Muthukumaraswamy, 2010*; *Brücke et al., 2012*). The Communication-through-Coherence theory describes how gamma synchrony can be a key feature of interregional interactions, with brief windows of increased excitability between longer periods of inhibition (*Hasenstaub et al., 2005*; *Fries et al., 2007*; *Buzsáki and Wang, 2012*) rendering neuronal

communication effective, precise and selective (*Fries, 2015*). One consequence of synchronized spiking activity is that where multiple cells converge onto a postsynaptic neuron, volleys of near-coincident spikes are more likely to trigger action potentials downstream (*Azouz and Gray, 2003*; *Harris et al., 2003*; *Gupta et al., 2016*). Neural synchronization could thus potentially trigger the currently unknown cascade of network interactions enabling movement execution. Uncontrolled movements recorded during levodopa-induced dyskinesia in Parkinson's disease are for example accompanied by excessive gamma synchrony between the STN and motor cortex (*Swann et al., 2016*). Gamma phase coupling between the STN LFP and motor cortex has also been shown during voluntary movements, with the former seemingly driving the latter (*Williams et al., 2002*; *Litvak et al., 2012*; *Sharott et al., 2018*). Moreover, dopaminergic medication enhances gamma synchrony in Parkinson's disease (*Litvak et al., 2012*), suggesting it may be functionally relevant for restoring these patients' ability to move.

Previously, it has been shown that changes in STN spike rhythmicity and phase coupling to cortical oscillations (focusing on frequencies below the gamma range) can occur before movement onset in the absence of changes in average firing rates (*Lipski et al., 2017b*). The precise temporal pattern of basal ganglia spikes relative to cortical population activity thus seems critical for understanding how the basal ganglia can contribute to motor control. Yet, studies on neural correlates of movement preparation and execution in the basal ganglia have mostly focussed only on changes in firing rates (*Anderson and Horak, 1985*; *Mink and Thach, 1991*; *Turner and Anderson, 1997*; *Seo et al., 2012*; *Arimura et al., 2013*). Hence not much is known about the relationship between gamma coupling and gating of activity for motor control, despite rapidly growing knowledge about the relevance of temporally organized activity for visual processing (*Fries, 2015*) and working memory encoding (*Bastos et al., 2018*).

We simultaneously recorded STN unit activity and electrocorticography from the motor cortex during isolated contra- or ipsilateral hand gripping movements to gain insights into mechanisms of cortico-basal ganglia communication that may be linked to gamma-rhythmic activity.

Inspired by previous findings that have demonstrated strong correlation between movement kinematics and basal ganglia gamma-band activity (*Brücke et al., 2012*; *Fischer et al., 2017*; *Lofredi et al., 2018*), we aim to shed light on the underlying neurophysiological mechanism resulting in cortico-subcortical communication in the gamma band. Therefore, we set out to examine if STN spike-to-cortical gamma phase coupling can predict the timing or vigour of contralateral action initiation. Second, because only contra- and not ipsilateral movements are accompanied by strongly increased motor cortical gamma synchrony (*Crone et al., 1998*; *Cheyne et al., 2008*) and result in more discharge activity (*Matsunami and Hamada, 1981*), we further characterized and compared the relationship between the precise timing of STN spikes and the cortical gamma phase. More specifically, if the timing of STN spikes helps to enhance gamma oscillations exclusively during contralateral gripping, then spikes during ipsilateral gripping may be clustered around points that are opposite to the preferred phase found during contralateral gripping to prevent a coincidental boost in gamma synchrony.

## Results

Cortical ECoG and STN single neuron spikes were recorded in Parkinson's disease patients undergoing deep brain stimulation surgery. Patients performed a visually-cued gripping task in which they squeezed a handgrip with their left or right hand within 2 s after the Go cue for at least 100 ms (*Figure 1*). Our analysis included 28 spike recordings from 12 patients (12 single-unit and 16 multi-unit recordings), from which spike-phase coupling to lower frequencies have been described previously (*Lipski et al., 2017b*).

### Gripping performance

Average reaction times from the Go signal to the grip onset were 0.53 ± (SD) 0.21 s and 0.55 ± 0.18 s for contra- and ipsilateral grips, respectively (contra-ipsi: p=0.435, $t_{27}$ = −0.8). The mean variability (SD) of RTs within each patient was 0.19 ± 0.09 s and 0.23 ± 0.12 s (contra-ipsi: p=0.050, $t_{27}$ = −2.1). This variability in RTs allowed us to compare the extent of spike-to-gamma phase coupling between trials with fast and slow RTs by grouping trials with RTs below the median and trials with RTs above the median. The average RTs of these two sets were 0.39 ± 0.16 s and 0.68 ± 0.27 s for grips

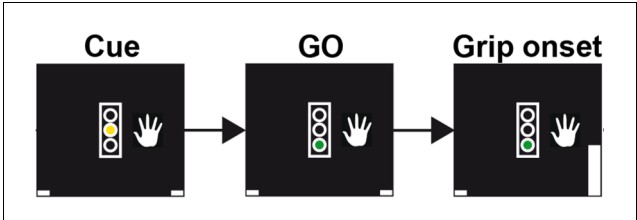

**Figure 1.** Schematic of the trial events for a grip trial with the right hand. In trials requesting gripping with the left hand, the image of the hand would be displayed to the left of the traffic light. The initial cue was displayed as yellow traffic light, which served as GO cue when it turned green, on average 1.5 ± 0.4 s after its onset. The gripping movement had to occur within two seconds of the GO cue onset to count as valid trial. Upon gripping, a white bar representing the grip force would move up (see bottom right corner in the rightmost plot). Visual feedback was presented according to whether the movement was performed at least for 100 ms and within the correct window. Trial onsets were on average separated by 4.7s ± 1.2s.

contralateral to the recorded STN and 0.38 ± 0.12 s and 0.72 ± 0.26 s for ipsilateral grips. The delay between the cue that instructed patients whether to perform the grip with the right or left hand and the Go signal did not differ significantly between short and long RT trials (difference in delay between long vs. short RT trials = 4 ms, p=0.116). We also performed median-splits according to grip peak force and peak yank (contralateral gripping peak force: 91 ± 65 N and 104 ± 71 N, peak yank: 0.39 ± 0.30 N/ms and 0.50 ± 0.36 N/ms). The average grip duration was 0.98 ± 0.45 s.

## Topography of STN spike-to-cortical gamma coupling

First, we assessed if movement-related STN spike-to-cortical gamma coupling was specific to a region of primary motor cortex that has also been demonstrated to show the strongest movement-related gamma power increase in MEG recordings *Figure 2B*, adapted from *Cheyne et al. (2008)*. This MEG study has shown a contralateral gamma increase at highly specific spatial locations for individual limb movements, which was highly consistent over repeated measurements and sessions (*Cheyne et al., 2008*; *Cheyne, 2013*).

ECoG strips were distributed predominantly over precentral and parietal areas and were always located in the same hemisphere as the recorded STN (*Figure 2A*). Because other studies have already clearly demonstrated the presence of finely tuned gamma activity and coupling in cortex and the STN (*Litvak et al., 2012*: Figures 5 and 6; *Cheyne et al., 2008*), we took a hypothesis-driven approach and focussed specifically on analysing STN spike coupling with 60–80 Hz cortical gamma activity in a 500 ms window around movement onset. Cortical power in the 12–30 Hz beta and the 60–80 Hz gamma frequency bands was modulated at movement onset (−0.1–0.4 s around grip onset) in bipolar contact pairs situated predominantly in primary motor and sensory areas (*Figure 2 C+D*), which is in line with past ECoG studies (*Ohara et al., 2000*; *Pfurtscheller et al., 2003*). Note that gamma power increased specifically for contralateral and not ipsilateral movements. Within-subjects correlations between the magnitude of the relative power changes in the beta and gamma band at contralateral movement onset did not significantly differ from zero at the group level, suggesting that they were uncorrelated with each other (mean R = 0.05, p=0.785, Wilcoxon signed-rank test, n = 28).

The bipolar contact pairs with the strongest cortical gamma phase coupling to STN spikes during contralateral gripping per spike recording were concentrated in lateral precentral gyrus (*Figure 2E*), a site, which corresponds strikingly well to the hotspot of 60–90 Hz gamma synchronization r in a previous MEG study that presumably identifies primary hand motor cortex *Figure 2B*, adapted from *Cheyne et al. (2008)*. The spatial focus of coupling during contralateral gripping becomes even clearer when only the sites are displayed for which movement-related STN-spike-to-cortical gamma phase coupling reached significance relative to a permutation distribution (*Figure 2F*). The sites that showed significant coupling during ipsilateral gripping were more widely dispersed and were shifted toward somatosensory cortex. The significance tests in *Figure 2F* show that gamma coupling was higher relative to a pre-movement period and relative to shuffled data, confirming that the increase

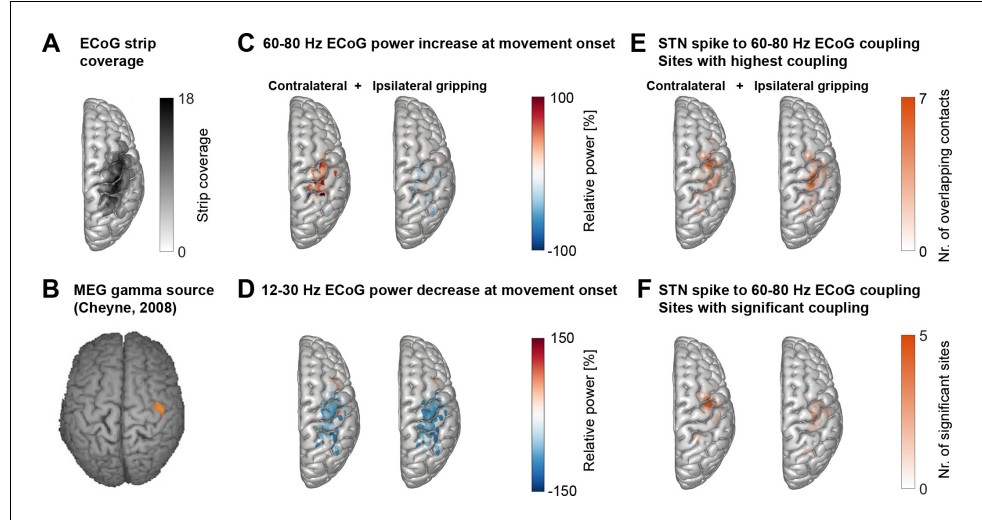

**Figure 2.** Cortical topography of STN spike-to-cortical gamma coupling and power changes. (**A**) ECoG strips were distributed mainly over precentral and parietal areas. ECoG coordinates from recordings performed in the left hemisphere were flipped in the x-axis to allow averaging across recordings performed in the left and right hemisphere. The maximum number of overlapping recordings was 18 (dark areas). (**B**) A magnetoencephalography study in healthy participants showed independently from our study that gamma oscillations at the onset of finger movements are focal to lateral motor cortex (*Cheyne et al., 2008*). (**C**) Gamma power increased most strongly over motor and somatosensory cortical areas during contralateral gripping. (**D**) Beta power decreased in spatially more widespread areas. (**E**) The left and right plots show the two sets of sites that showed the highest 60–80 Hz STN spike-to-cortical gamma coupling during contralateral and ipsilateral gripping, respectively. Each set includes one single site per recorded hemisphere (n = 28). These sites were chosen for further analyses. (**F**) As (**E**) but reduced to the sites showing significant gamma coupling −0.1–0.4 s around movement onset (α = 0.1 to show the location of a larger number of contacts). If several channels of one recording were significant, only the channel with the highest PLV was included. The contacts that showed significant gamma PLV during contralateral gripping were concentrated over precentral cortex. This hotspot corresponds well with the 60–90 Hz gamma source localized in MEG studies (*Cheyne et al., 2008*) shown in **B**). The significant contacts for ipsilateral gripping were scattered over a wider area. See also *Figure 2—figure supplement 1*.

© 2008 Elsevier. *Figure 2B* is reprinted with permission from *Cheyne et al. (2008)* . It is not covered by the CC-BY 4.0 licence and further reproduction of this panel would need permission from the copyright holder.

The online version of this article includes the following figure supplement(s) for figure 2:

**Figure supplement 1.** Distribution of coupling strength across all ECoG sites in one example subject recorded with a grid that had 28 contacts.

---

exclusively occurred at movement onset and was not already present at rest. Sites with the highest coupling during contralateral and ipsilateral gripping were separately selected for further analyses.

## Movement-related STN spike-to-cortical gamma coupling

To evaluate changes in coupling across the whole task period and across multiple frequencies at the group-level, we employed a novel procedure that ensured that the time-frequency plots depict changes in the data locked to each task event. Each trial began with a Left/Right cue that was followed by a GO signal, movement onset and movement offset (*Figure 1*). To calculate coupling strength for multiple time points and make sure the values are comparable, the number of spikes going into the calculation should be kept constant for each point because smaller numbers would result in inflated coupling values. Intervals between events were divided into equidistant points for each trial so that the variability of these intervals across trials and subjects did not affect the accuracy of the data alignment around important task events. The windows centred around these points were scaled such that each window contained the same number of spikes to then compute the phase locking value (PLV) for this point (see Methods). Movement-related cortical power changes showed a broad power increase between 60–150 Hz (*Figure 3B*) that is typically observed in ECoG recordings. Such broad power increases can reflect both true oscillatory components as well as non-oscillatory

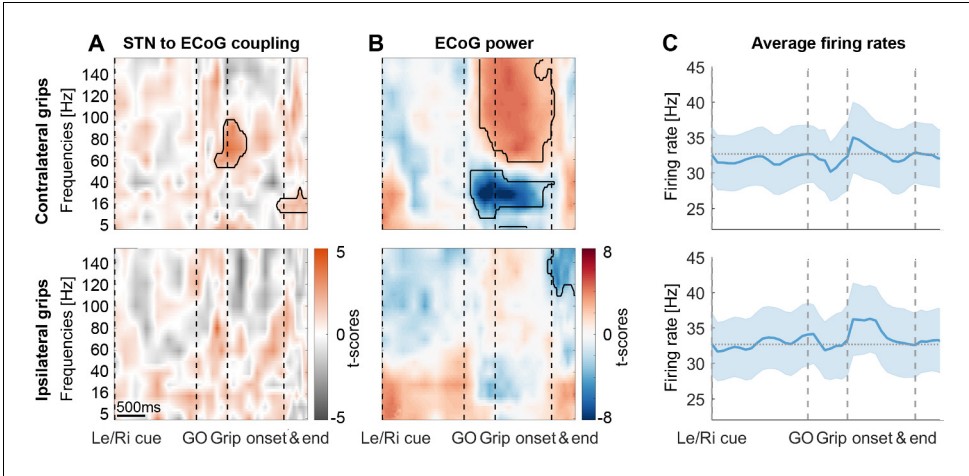

**Figure 3.** STN spike-to-ECoG phase coupling and ECoG power during visually cued gripping. The top and bottom row show contra- and ipsilateral grip trials, respectively. To get comparable estimates of the coupling strength around each of the key trial events, we implemented an event-locked variable-window width PLV estimation procedure: Each interval between neighbouring task events was subdivided into equidistant points, which served as centre for a window within which the PLV was computed. The windows varied in width to encompass the same number of spikes for all time points for one recording. (A) The black outlines show significant clusters obtained with a cluster-based permutation procedure for multiple comparison correction (n = 28, p<0.05). Significant 60–80 Hz spike-to-cortical gamma phase coupling occurred at contralateral grip onset. The same cells also locked to beta after movement offset when cortical beta power rebounded. During ipsilateral gripping (bottom row), no significant clusters were found. (B) During contralateral gripping, cortical gamma power increased while beta power was significantly suppressed relative to the median power obtained from the whole recording. Ipsilateral gripping resulted in relatively reduced >120 Hz power after the grip was released. (C) Average firing rates across the 28 recordings showed no significant modulation to any of the task events. Error bars display standard errors. See also *Figure 3—figure supplements 1–5*.

The online version of this article includes the following source data and figure supplement(s) for figure 3:

**Source data 1.** Source data for *Figure 3* showing contralateral grip trials.
**Source data 2.** Source data for *Figure 3* showing ipsilateral grip trials.
**Figure supplement 1.** Example of the event-locked variable-window width sliding procedure.
**Figure supplement 2.** Spike-triggered average (STA) of the 60–80 Hz filtered ECoG signal during contralateral gripping of three example recordings.
**Figure supplement 3.** Examples of cells that are coupled to cortical gamma at contralateral movement onset and either maintain stable firing rates (top row), show a mixed response (middle right), increase (bottom right) or decrease (bottom left) their firing rates.
**Figure supplement 4.** Additional firing rate analysse.
**Figure supplement 5.** During ipsilateral gripping, STN spikes can be locked to gamma recorded from different cortical sites.
**Figure supplement 5—source data 1.** Source data to *Figure 3—figure supplement 5* showing coupling with IPSI GRIP ECoG sites during contralateral grip trials.
**Figure supplement 5—source data 2.** Source data to *Figure 3—figure supplement 5* showing coupling with IPSI GRIP ECoG sites during ipsilateral grip trials.

broadband activity capturing short neural events (*Ray and Maunsell, 2011*). To focus on the 60–80 Hz oscillatory component of interest, we pre-selected the ECoG contacts with the highest PLV between 60–80 Hz over a −0.1–0.4 s period around movement onset (*Figure 2E*). Accordingly, STN firing was consistently phase-locked to cortical gamma at movement onset in these sites across all recordings (*Figure 3A*, n = 28). The significant cluster shows that coupling strength was higher than that obtained from a permutation distribution, which was generated by shuffling the association between STN spike trains and cortical LFP gamma phase time courses across trials, while keeping the respective individual trials intact. The cluster highlights that in the selected sites, above-chance coupling was relatively constrained to a narrow spectral window in the gamma band and centred narrowly in time on movement onset instead of being widespread, which would have been possible

in principle, despite the selection criterion. *Figure 3—figure supplement 2* shows the spike-triggered average of the 60–80 Hz filtered ECoG signal of three example recordings demonstrating clear locking around contralateral grip onset but not in a pre-movement period. During ipsilateral gripping, no significant clusters in coupling between STN spikes and the gamma phase of this preselected set of ECoG sites were found (*Figure 3A*, bottom row) and no significant gamma increase was present (*Figure 3B*). The cortical high-frequency activity (HFA, described in Methods) as a proxy of local discharge activity was also significantly higher during contra- versus ipsilateral gripping (Wilcoxon signed-rank test, n = 28, p=0.001).

The average firing rate at the group level did not change with any of the task events (*Figure 3C*). This was the case not only when computing average firing rates in 0.3 s long sliding windows as shown in *Figure 3C*, matching the average window length used to calculate the coupling strength, but also when examining instantaneous firing rates.

Importantly, gamma coupling could be found in recordings of all response types, irrespective of whether firing rates remained stable (47%), increased (21%) or decreased (32%) (*Figure 3—figure supplement 3*, cells were classified as increasing or decreasing if a significant cluster occurred between the Go cue and movement offset; some additionally showed a significant decrease either before or after the movement). We evaluated firing relationships further by computing a correlation across recordings between firing rate changes at movement onset and coupling strength, which confirmed that no relationship was present (mean R = 0.01, p=0.969, see *Figure 3—figure supplement 4A*). The only cue-related change in firing characteristics observed on the group level was a reduction in the interspike interval coefficient of variation (*Figure 3—figure supplement 4B*). In summary, these results show that spatially and temporally specific STN-spike-to-cortical-gamma coupling can be detected at the onset of contralateral movement.

We then extended our analyses to ECoG sites that showed the greatest spike-to-gamma coupling during ipsilateral gripping. In this set of sites, coupling also exceeded that obtained by a permutation distribution (*Figure 3—figure supplement 5B* bottom). Note that these sites were spatially more widely distributed and tended to be located posterior to motor cortex (*Figure 2E* right, labelled IPSI GRIP ECoG sites). In these sites, gamma coupling was also present during contralateral gripping (*Figure 3—figure supplement 5B* top), although the coupling frequency appears to be slightly higher than for the set of ECoG sites in which coupling was greatest for contralateral gripping. Gamma power increased and beta power decreased significantly after cluster-based permutation correction relative to the median power obtained from the whole recording only for contralateral grip trials (*Figure 3—figure supplement 5C*). Notably, if the initial cue indicated an ipsilateral grip, beta power was significantly elevated until the Go signal while power above 60 Hz was significantly reduced (*Figure 3—figure supplement 5C*). Next, we tested if the coupling strength preceding the movement onset related to reaction times.

## Stronger spike-to-gamma coupling precedes faster reaction times

To test if reaction time differences were linked to differences in gamma coupling, we median-split all trials and calculated coupling strengths separately for the two subsets. This analysis revealed that faster reaction times were preceded by higher STN spike-to-primary motor cortical LFP gamma coupling and that this was specific for contralateral gripping. The difference was present already at the time of the Go signal (*Figure 4A*), which occurred on average 0.39 s before grip onset in fast trials and 0.69 s in trials with slow RTs. The windows for calculating PLVs spanned on average −0.16:0.16 s around each point and the precise window length at the Go signal was 306 ± 68 ms and 310 ± 75 ms for fast and slow RT trials. Thus, even in trials with short RTs, the movement onset fell outside the window used for calculating PLVs at the time of the Go signal, where a significant difference in coupling already was present ($t_{27}$ = 3.2, p=0.004). Hence, the difference in coupling cannot be explained merely by earlier movements in trials with short RTs. The difference disappeared at grip onset, which shows that the coupling strength during the movement itself was comparable irrespective of whether RTs leading up to the movement were fast or slow. To further corroborate this finding, we also computed within-subjects correlations between binned coupling strengths and RTs in a 0.5 s long window starting at the Go cue. This confirmed again at the group level that RTs were significantly shorter when coupling strength was higher (mean R = −0.16, $t_{27}$ = −2.5, p=0.019).

No such correlation was present between RTs and firing rates (mean R = −0.02, $t_{27}$ = −0.5, p=0.594) and none of the firing characteristics differed between trials with fast and slow RTs

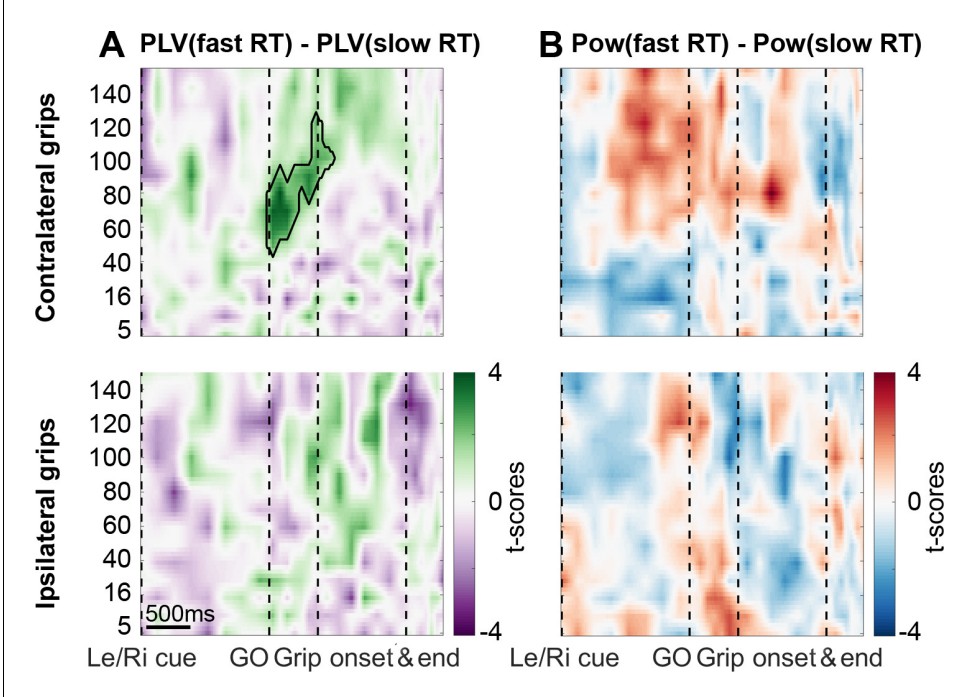

**Figure 4.** Coupling was significantly higher when reaction times were faster. (**A**) The green cluster shows that in those trials, in which RTs were fast as defined by a median split, gamma coupling in the contralateral hemisphere was already significantly higher immediately after the Go signal, about 500 ms before movement onset. (**B**) Average cortical gamma power tended to be higher and pre-Go signal beta power lower in contralateral grip trials with faster reaction times, however, these differences did not survive cluster-based multiple comparison correction. See also *Figure 4—figure supplements 1* and *2*.

The online version of this article includes the following figure supplement(s) for figure 4:

**Figure supplement 1.** No significant differences in firing characteristics were found when comparing trials with fast vs. slow RTs.

**Figure supplement 2.** Analyses related to yank and force.

(*Figure 4—figure supplement 1*). When trials were split according to peak yank or peak grip force, no significant effects were found (*Figure 4—figure supplement 2*). Note that primary motor cortical gamma power also tended to be higher when reaction times were shorter, although the effect was not significant (*Figure 4B*). Finally, we also found that the RT-dependent difference in coupling was spatially specific, as it was not present in ECoG sites that showed the greatest coupling during ipsilateral gripping.

## STN spiking probability is modulated relative to the cortical gamma phase

PLVs for individual pairs of STN spike and cortical ECoG recordings as a measure of coupling strength do not provide a measure of the consistency of the preferred phase across recordings. As only the length of the average phase vector is used to calculate PLVs, reflecting how strongly bundled or spread the phase values are, information about the preferred phase is lost. To find out whether, across recordings, spikes were consistently more probable at certain points of the gamma cycle and less probable at others, we investigated the spiking probability relative to the cortical gamma phase. It would be unlikely to find a consistent phase preference and significant modulation of spiking probabilities if high PLV coupling was a by-product of analysis choices (e.g., the pre-selection of ECoG contacts).

Spiking probabilities were computed within a −0.1–0.4 s window around movement onset for four non-overlapping gamma phase bins. To test whether the resulting probabilities were significantly modulated, we computed a 4 (*bins*) x 2 (*effector side: contralateral vs. ipsilateral*) ANOVA.

The ANOVA resulted in a significant interaction ($F_{3, 81}$ = 5.2, p=0.003) and no significant main effects (factor bins: $F_{3, 81}$ = 2.4, p=0.090; factor effector side: $F_{1, 27}$ = 0.1, p=0.718). *Figure 5* shows that spiking probabilities were significantly and differentially modulated during contralateral (BIN2 vs. BIN4) and ipsilateral gripping (BIN1 vs. BIN2/3/4). This significant modulation shows that the preferred and non-preferred phases for spiking were relatively consistent across recordings, despite the ECoG sites used to extract the gamma phase were selected according to the highest coupling irrespective of the preferred phase.

The spiking probability in BIN4 was significantly lower during contralateral compared to ipsilateral gripping (Wilcoxon signed-rank test, n = 28, p=0.008) and the probability in BIN1 was higher ($t_{27}$ = 2.5, p=0.021, not significant if corrected for the four multiple comparisons). Increasing the number of bins to five instead of four resulted in a similar significant modulation within contra- and ipsilateral grip trials (*Figure 5—figure supplement 3*), however, none of the bins showed a significant difference between contra- and ipsilateral gripping with this subdivision. Only the ANOVA or the pairwise comparisons for gamma obtained from primary motor cortical ECoG sites were significant. Spiking probabilities for gamma obtained from the other set of sites (with highest coupling during ipsilateral gripping) are shown in *Figure 5—figure supplement 4*.

This distinct modulation of spiking probability according to the phase of gamma is remarkable and may provide a mechanism that enhances cortical gamma oscillations, which were more pronounced during contralateral gripping (*Figure 3*). If STN spikes that occur at one phase of the cortical gamma cycle contribute to boosting cortical gamma for contralateral movements, then during ipsilateral movements unintentional boosting of gamma could be avoided if spikes would occur at the opposite phase (i.e. 180° apart). The following section will quantify if such a systematic difference in spike timing was present. This analysis was also motivated by the above reported differences in spiking probabilities between contra- and ipsilateral gripping observed at a specific phase of the gamma cycle.

### The preferred gamma phase of STN spikes differs between contra- and ipsilateral movements

To test if the timing of STN spikes relative to the cortical gamma phase systematically differed between contra- and ipsilateral grip trials, we extracted the gamma phase from the primary motor cortical ECoG sites and computed the average, or 'preferred', phase coinciding with STN spikes for both trial types. To exclude recordings where estimation of the average phase estimate may have been unreliable, we started conservatively by using only recordings in which gamma coupling was significant at movement onset (shown in *Figure 2F*). We found in this subset (n = 11) that the preferred gamma phase differed significantly between contra- and ipsilateral grip trials by ~ 210° (mean offset −2.6 rad, 95% CI = [−3.9,−1.4], *Figure 6A*). A more relaxed selection included all recordings in which the pairwise phase consistency value (a measure of unbiased coupling strength) exceeded zero (see Methods). In the larger set (n = 23), which is potentially more representative although individual phase estimates may be noisier, the preferred phase was offset by approximately 180° (*Figure 6B*, mean offset 3.1 rad, 95% CI = [2.1, 4.0]). P-values derived from a V-test assessing directionality towards an offset of 180° showed that, as expected, the significant phase difference was only present around movement onset (*Figure 6C*), where coupling was strongest. This analysis thus confirmed that spikes preferentially locked to approximately opposite points of the gamma cycle depending on whether the selected motor response was contralateral or ipsilateral to the recorded hemisphere.

### Discussion

We performed simultaneous recordings of STN unit activity and electrocorticography during DBS surgery while subjects performed a hand grip task. We found that enhanced STN spike-to-cortical gamma phase coupling preceded faster RTs already at the time of the Go cue, suggesting a role for subcortico-cortical coupling in preparing movement. This effect was spatially specific to ECoG sites that were predominantly located over primary motor cortical areas. Importantly, we found no correlations between reaction times and firing rates in the same time window, indicating that the timing of neuronal discharges with respect to cortical gamma phase was more predictive of movement than

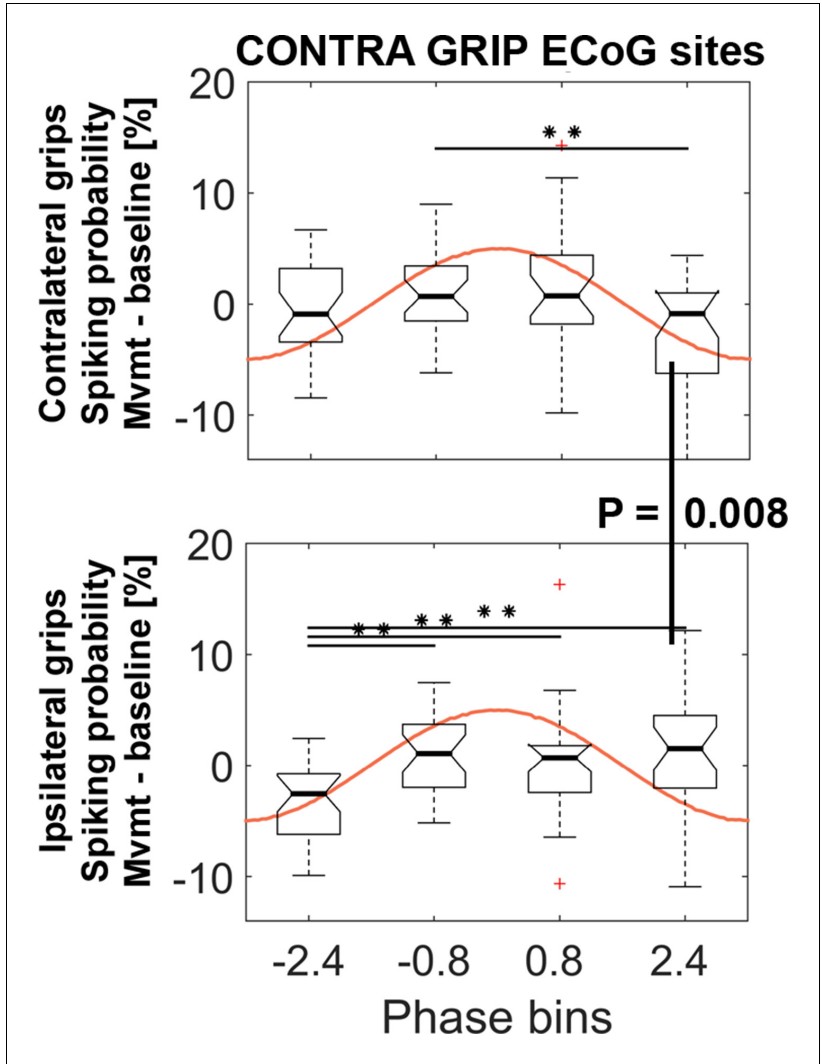

**Figure 5.** STN spiking probability is modulated relative to the phase of cortical gamma. The top plot shows how spiking probabilities during contralateral grip onset co-modulate with the gamma phase. * show FDR-corrected significant differences between phase bins. Spiking was relatively reduced in BIN4 during contralateral gripping. It was significantly lower than during ipsilateral gripping (bottom plot, Wilcoxon signed-rank test, p=0.008). The central mark of the box plots displays the median and edges show the 25th and 75th percentile. Whiskers show the 1.5*interquartile range and outliers (red crosses) are data points beyond this range. See also *Figure 5—figure supplements 1–4*.

The online version of this article includes the following figure supplement(s) for figure 5:

**Figure supplement 1.** Description of the rationale behind the polarity flipping procedure.

**Figure supplement 2.** Polarity standardization procedure.

**Figure supplement 3.** Similar analyses as in *Figure 5* based on the same ECoG sites but with five instead of four phase bins.

**Figure supplement 4.** Similar analyses as in *Figure 5* but based on gamma obtained from the set of ECoG sites that showed the strongest coupling during ipsilateral gripping.

firing rates. *Table 1* shows an overview of the set of questions on STN spike-to-cortical gamma phase coupling that we covered in this study.

We also showed that STN spiking probability was significantly modulated when binned with respect to the gamma cycle. This is a strict test of the temporal consistency of coupling across recordings because it considers the bin preferences of each of the recorded spike trains.

The STN receives widespread glutamatergic cortical inputs and has reciprocal connections with the GABAergic globus pallidus externus (GPe) (*Albin et al., 1989*). This reciprocal connectivity may

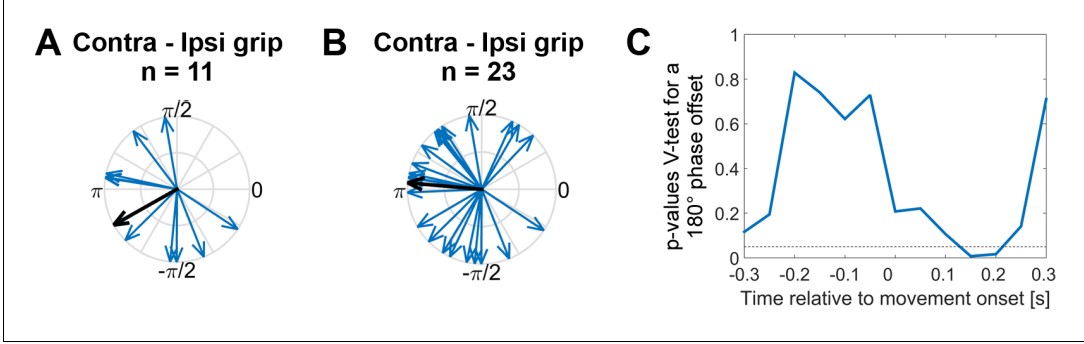

**Figure 6.** Systematic offset of STN spike timing relative to cortical gamma between contra- and ipsilateral gripping. (**A**) STN spikes of recordings with significant gamma coupling (n = 11 units from seven patients [n = 6: one unit, n = 1: five units]) coincided with cortical gamma phases that were on average nearly opposite when comparing contra- and ipsilateral gripping (mean offset = −2.6 rad, 95% CI = [−3.9,−1.4]). (**B**) When including more recordings (all recordings where PPC > 0, n = 23 from 12 patients [n = 7: one unit, n = 1: two units, n = 3: three units, n = 1: five units]), the offset was very close to 180° (mean offset = 3.1 rad, 95% CI = [2.1, 4.0]). The gamma phase was extracted from ECoG sites that showed the highest coupling during contralateral gripping, as also used for *Figures 3–5*. (**C**) P-values derived from a V-test assessing directionality towards an offset of 180° (n = 23) show that the effect of the offset is strongest in a 500 ms sliding window centred around 0.15 s after movement onset, i.e. in a window −0.1–0.4 s around movement onset.

in itself be able to generate, boost or sustain rhythmic activity (*Blenkinsop et al., 2017*). Considering that STN activity strongly affects the thalamus by exciting the inhibitory basal ganglia output nuclei (GPi/SNr), gamma-rhythmic STN firing could enhance cortical gamma oscillations by switching the GPi from a tonic inhibitory mode to an oscillating mode with brief periods of silence that then disinhibit the thalamus. Note that in the present study, we could not determine whether coupling and local gamma synchrony originated within the basal ganglia or in cortex, which directly projects to the STN via the hyperdirect pathway. Although previous studies suggest that STN gamma activity drives that in cortex (*Williams et al., 2002*; *Litvak et al., 2012*; *Sharott et al., 2018*), coupling may well occur through transient cortico-STN-cortical network effects.

Importantly, average STN firing rates across all recordings showed no consistent task-related change. We only found an increase in the regularity of spike timing, which was expressed as decrease in interspike interval variability after the preparatory cue and in ipsilateral trials even during gripping. Rhythmicity of spiking itself at this fast temporal scale and in short windows can be hard to detect in individual cells with autocorrelation measures because sparse spiking makes it difficult to capture a clear oscillatory structure.

Further support for the hypothesis that the relative timing of STN spikes matters for selectively executing contra- or ipsilateral actions was provided by our finding that the average gamma phase that coincided with STN spikes was systematically offset between contra- and ipsilateral gripping: During ipsilateral gripping, the timing of STN spikes was shifted by ~ 180° relative to the preferred

**Table 1.** Questions addressed by STN spike-to-cortical gamma phase coupling analyses.

| Figure | Question | Finding |
|---|---|---|
| *Figure 2*: | Is it most pronounced over motor cortex? | Yes |
| *Figure 3A*: | Is it frequency- and time-specific? | Yes |
| *Figure 3C*: | Is the average firing rate across all units significantly modulated? | No |
| *Figure 4*: | Are faster reaction times linked to stronger coupling? | Yes |
| *Figure 5*: | Is the probability of spikes significantly modulated relative to the cortical gamma phase | |
| | during contra- and ipsilateral gripping? | Yes |
| *Figure 6*: | Do we find a systematic phase offset of the average timing of STN spikes relative | |
| | to the cortical gamma phase between contra- and ipsilateral gripping? | Yes |

phase during contralateral gripping. The preferred phases thus were located at approximately opposite points of the gamma cycle. Presuming that the gamma cycle captures alternating phases of relative depolarisation and hyperpolarisation of a large number of cortical cells, this may be a mechanism to dynamically modify the functional consequences of STN spikes. Neurons in the GPi receive excitatory afferents from the STN but also inhibitory afferents from the GPe (*Smith et al., 1994*) and the striatum (*Albin et al., 1989*), which is also under direct cortical influence and may thus also be gamma-entrained. Depending on the timing of converging cortical activity coming through the STN and the striatum, gamma synchrony in the basal ganglia could be enhanced or suppressed. For instance, if STN spikes are focussed within brief windows, strong inhibitory striatal inputs may coincide with them in the GPi and thus diminish the overall excitatory drive of the STN, even if STN spike rates remain the same. Reduced GPi firing could then promote thalamic and cortical facilitation – correlates of which we saw as an increase in HFA during contralateral gripping. Conversely, during ipsilateral gripping, if the relative timing of STN spikes is offset by ~ 180°, these spikes may drive GPi cells outside of periods of incoming inhibition. This drive could maintain GPi firing and thus maintain tonic inhibition of the thalamus as would be appropriate when initiating ipsilateral movements. Also at the level of the thalamus, the relative timing of incoming activity should matter substantially as it integrates excitatory cortical and inhibitory basal ganglia inputs (*Goldberg and Fee, 2012*). All of these speculations would be tested ideally in future studies with simultaneous recordings in the relevant sites.

Although many specifics of the present results are unique, recent publications support the general idea that shifts in the relative timing of neuronal activity in different sites of the cortico-basal ganglia-thalamo-cortical loop may have a strong influence on the function of the circuit (*Goldberg and Fee, 2012*; *Jantz et al., 2017*). Precise temporal patterning of STN spikes could also provide a key mechanism to avoid co-activation of cells that may otherwise activate competing effectors during movement. Losing control over the mechanisms that enable precise temporal structuring of activity might therefore lead to uncontrolled movements as seen in Huntington's disease or levodopa-induced dyskinesia.

Because ECoG signals tend to show a very broad movement-related power increase between 50–200 Hz, past research has primarily focussed on the coupling of spikes to the amplitude of 50–200 Hz activity (*Shimamoto et al., 2013*; *Lipski et al., 2017b*), which captures aspects of neural activity that are separate from the narrow-band gamma phase. The movement-related increase in ECoG power in our data also spanned a very broad frequency range from 60 to 150 Hz. It is widely acknowledged that ECoG activity in this range can contain both true oscillatory components but also broadband activity that is devoid of rhythmicity (*Ray and Maunsell, 2011*; *Bonnefond et al., 2017*). Distinguishing the two is difficult, because true synchronization between distinct sites can be present even despite the absence of a clear peak in the power spectrum (*Vinck et al., 2013*; *Brunet et al., 2014*). In our data, we assume that the broadband power increase originated partly due to increased neural activity, including postsynaptic potentials and action potentials, but that the phase of the 60–80 Hz filtered signal is still meaningful, because discrete spectral peaks in this band have been reported in MEG recordings of healthy participants and patients with PD, demonstrating the presence of pronounced neural synchrony (*Litvak et al., 2012*; *Cheyne et al., 2008*). Note that in ECoG sites that were further away from the presumed hand region of primary motor cortex, the relative contribution of broadband neural activity may be larger and the oscillatory component may be weaker. Our choice of extracting the 60–80 Hz phase also appears valid when considering the clear co-modulation of the amplitude of high-frequency activity with the 60–80 Hz filtered signal. The phase of 60–80 Hz gamma activity thus seems to reflect when synaptic inputs arrive in volleys that may then trigger action potentials captured as high-frequency activity (*Fries et al., 2007*; *Buzsáki et al., 2012*). As expected, we found that cortical HFA was higher during contralateral gripping.

We also observed that the same cells that dynamically locked to gamma activity can likewise lock to beta oscillations after movement completion. A rebound of post-movement beta oscillations has been associated with evaluation of a movement outcome. When feedback is as expected, the beta rebound is known to be high (*Tan et al., 2014*; *Tan et al., 2016*; *Torrecillos et al., 2015*). High spike coupling to beta thus may consolidate the previous movement or bias against adaptation of the next motor response, although this remains to be tested. In this regard it is also interesting that if the initial cue indicated an ipsilateral grip, pre-movement cortical beta power was significantly

elevated throughout the pre-Go signal delay period, consistent with an overall anti-kinetic influence on contralateral output units prior to ipsilateral gripping. The variability of interspike intervals simultaneously decreased. Our findings support the emerging picture of two (although there may be more) distinct rhythms, the beta and the gamma rhythm, that can entrain neuronal spikes at different stages of motor control to facilitate or restrict motor output. In this light, gamma oscillations may reflect processing that is necessary for a change in motor state, such as the start of a movement or sudden cessation (*Fischer et al., 2017*). The exact downstream consequences of STN spike coupling to different rhythms remain to be investigated with multi-site recordings in strictly controlled motor tasks.

Differences in gamma coupling were linked to fluctuations in reaction times, but not in movement velocity or force. This seems at first surprising as local STN gamma power has been shown to correlate with movement velocity (*Joundi et al., 2012*; *Lofredi et al., 2018*) and force (*Tan et al., 2013*; *Alhourani et al., 2018*). However, our task was not designed to capture a wide range of grip onset velocities or force levels and patients were studied during dopamine withdrawal, which reduces velocity-related gamma modulation (*Lofredi et al., 2018*). Because patients were not pressed to move as fast as possible, reaction times varied considerably, which allowed us to examine how coupling strength differed despite not specifically manipulating motor preparation experimentally. A similar relationship, but at an across-subjects correlation level, between faster reaction times and enhanced STN LFP-to-motor cortex gamma coupling was recently reported in a dataset with overlapping subjects (*Alhourani et al., 2018*). This study also reported a positive correlation between local STN gamma activity and the peak force. Local STN synchrony thus seems to be a stronger correlate of movement vigour than long-range STN spike-to-cortical gamma synchrony.

Note that because coupling strength varies strongly with the nature of the captured units, and thus different units recorded from one patient can result in variable coupling values, we did not perform correlations between coupling strength and disease severity. To meaningfully compute such correlations, the number of cells sampled from each patient would need to be large enough to compute a reliable mean coupling score.

Our study is limited in that ECoG coverage was not the same across patients and spanned limited areas of the cortical surface. However, we were able to perform group analyses by focussing on two sets of sites that showed the highest gamma coupling with contralateral and ipsilateral gripping. We found spatially specific effects, such as the difference in coupling depending on reaction times and the phase-dependent spiking probability modulation. These contrasts were unrelated to and did not depend on the ECoG contact pre-selection criterion and hence confirm that the selection procedure resulted in a physiologically relevant subset of ECoG sites involved in motor processing. The sites showing the highest gamma coupling during contralateral gripping were predominantly located precentral, in an area of motor cortex that seems to be involved in the control of upper limb movements (*Cheyne et al., 2008*). One site was located in lateral parietal area 5, which is also associated with direct corticospinal control of hand movements (*Rathelot et al., 2017*) and we cannot exclude that coupling may also be present between STN spikes and other cortical sites that were not recorded. We acknowledge that electrode localizations were determined indirectly, as our intraoperative imaging modality was two dimensional, however our two-dimensional to three-dimensional fusion technique previously was shown to localize ECoG recording locations with high functional-anatomic accuracy (*Randazzo et al., 2016*). Another caveat in this study is that firing rates in the STN were obtained from Parkinson's disease patients. Firing rates in patients at rest increase on average as the disease progresses (*Remple et al., 2011*). Whether our findings translate to neurologically healthy motor networks remains to be tested, although kinematics during task performance did not differ from those of subjects without a movement disorder (*Kondylis et al., 2016*). Note that according to the hemibody upper limb UPDRS scores reported in *Table 2*, symptom severity was on average worse on the ipsilateral side, which could also contribute to the difference seen between contralateral and ipsilateral gripping. The coupling that we observed also may be less specific than in healthy brains considering that progressive cell death in Parkinson's disease has been associated with a loss of effector-specific activity (*Bronfeld and Bar-Gad, 2011*). Note that we included both single- and multi-unit recordings in the analysis. Considering that multiple cells need to be recruited in rhythmic firing to cause any downstream effects, inclusion of multi-unit recordings may even be beneficial when investigating spike-to-gamma phase coupling. Finally, our results are correlative, and we

**Table 2.** Clinical details.

Age is given in years, M/F = male/female, MMSE = Mini Mental State Exam for neurocognition, UPDRS = Unified Parkinson's Disease Rating Scale, contra = contralateral, ipsi = ipsilateral, RAM = rapid alternating movements of the hands, Recorded HS = Recorded Hemisphere, NR = value was not recorded in the medical record.

| Age | Sex | MMSE Score | Tremor Dominant | Rest tremor R | Rest tremor L | Postural tremor of hands R | Postural tremor of hands L | Finger tapping R | Finger tapping L | Hand movement R | Hand movement L | RAM R | RAM L | Total UPDRS | Handedness | Recorded HS | Higher UPDRS | Nr. of units: contralateral + Ipsilateral grip trials |
|---|---|---|---|---|---|---|---|---|---|---|---|---|---|---|---|---|---|---|
| 69 | F | 29 | No | 0 | 2 | 0 | 0 | 1 | 1 | 1 | 1 | 1 | 1 | 31 | left | left+right | contra+ipsi | n = 2: 31, 27 |
| 54 | F | 29 | NR | NR | NR | NR | NR | NR | NR | NR | NR | NR | NR | NR | right | left+right | NR | n = 3: 43, 42 |
| 68 | M | 29 | No | 0 | 1 | 0 | 1 | 2 | 3 | 2 | 3 | 1 | 3 | 42 | right | left | ipsi | n = 1: 10, 16 |
| 66 | M | 30 | Yes | 3 | 3 | 2 | 3 | 3 | 3 | 3 | 3 | 3 | 2 | 62 | right | left+right | equal | n = 3: 68, 81 |
| 71 | M | 30 | No | 2 | 0 | 0 | 0 | 3 | 2 | 3 | 3 | 3 | 3 | 55 | right | left+right | contra+ipsi | n = 3: 61, 58 |
| 65 | F | 26 | No | 0 | 1 | 1 | 1 | 2 | 4 | 2 | 3 | 2 | 2 | 54 | NR | left | ipsi | n = 5: 96, 63 |
| 45 | M | 29 | Yes | 3 | 3 | 4 | 4 | 0 | 1 | 0 | 1 | 0 | 1 | 31 | right | left | ipsi | n = 1: 12, 12 |
| 60 | M | 29 | No | 2 | 3 | 1 | 1 | 2 | 3 | 2 | 2 | 2 | 2 | 52 | right | left | ipsi | n = 2: 42, 50 |
| 54 | M | 39 | No | 1 | 0 | 0 | 0 | 2 | 2 | 2 | 2 | 3 | 3 | 33 | right | left | contra | n = 1: 15, 13 |
| 68 | M | NR | No | 0 | 0 | 2 | 2 | 2 | 3 | 2 | 2 | 2 | 3 | 48 | right | left | ipsi | n = 1: 12, 11 |
| 52 | M | NR | No | 0 | 0 | 0 | 0 | 1 | 1 | 1 | 1 | 1 | 1 | 27 | right | left+right | equal | n = 5: 113, 112 |
| 60 | M | 30 | Yes | 2 | 2 | 1 | 2 | 1 | 1 | 1 | 1 | 1 | 1 | 28 | right | right | contra | n = 1: 16, 16 |

cannot infer whether any electrophysiological correlates were actually causal for an observed behaviour, such as faster reaction times.

Taken together, our work provides intriguing evidence for the idea that the temporal structure of activity travelling through the basal ganglia is important for flexible motor control. Further investigation of information routing principles with multi-site recordings in the basal ganglia in simple movement tasks may provide essential new insights about basic mechanisms of neural communication for movement initiation. Such insights could possibly transform our ability to improve treatments for basal ganglia disorders and our understanding of adaptive motor control.

## Materials and methods

### Subject details

A subset of the data recorded for this study was previously published elsewhere (*Lipski et al., 2017b*). 12 Parkinson's disease patients performed a visually cued hand gripping task while undergoing deep brain stimulation surgery after overnight withdrawal from or a reduced dose of their dopaminergic medication. They provided written, informed consent in accordance with a protocol approved by the Institutional Review Board of the University of Pittsburgh (IRB Protocol no. PRO13110420). Demographic details are reported in *Table 2*.

### Method details

Task

The behavioural task was previously described (*Kondylis et al., 2016*; *Lipski et al., 2017b*). Patients had to perform unimanual left- or right-hand grip movements for at least 100 ms with at least 10% of their previously determined maximal grip force. Considering that patients were currently undergoing deep brain stimulation surgery, they were not incentivised to respond as quickly as they could. Only trials with RTs longer than two seconds were treated as invalid. A single trial started with a yellow traffic light and an instructional cue either to the left or the right of it on a screen (*Figure 1*). The cue instructed participants whether to grip with the left or right hand. After a random interval ranging between 1–2 s, the yellow light disappeared and a green or a red light came on. The green light instructed the patients to move and the red light served as NOGO cue and instructed patients to withhold the movement. The number of NOGO trials was variable across patients and not all patients performed a sufficient number of NOGO trials, thus NOGO trials were not analysed. The Go signal was presented on average 1.5 ± 0.4 s after the instructional cue. After trial completion, visual feedback was provided in the form of 'You won $10' (or in a subset of patients 'You won $1') if the movement was performed correctly or 'No Gain' if it was performed incorrectly to keep patients engaged. The feedback was presented 1.9 ± 0.4 s after the Go signal and on average 0.43 ± 0.41 s after the grip was released. The average interval between the feedback and the next instructional cue was 1.4 ± 0.3 s (variable interval between 1–2 s), so that on average every 4.7s ± 1.2s a new trial started.

Electrophysiological recordings

STN microelectrode recordings were obtained with single glass-insulated platinum-iridium microelectrodes (FHC, Bowdoin ME) with impedances between 0.3 and 0.9 MΩ. Signals were filtered and amplified using the Guideline 4000 LP+ system (FHC; 125 Hz–20 kHz) and digitized at 30 kHz using the Grapevine Neural Interface System (NIPS; Ripple, Salt Lake City, UT). In one subject, the recordings were carried out using the Neuro-Omega recording system (Alpha Omega, Alpharetta, GA) using Parylene-insulated tungsten microelectrodes. Electrocorticography (ECoG) data were concurrently recorded with subdural strip electrodes placed temporarily via the existing burr hole used for DBS lead placement. The strips consisted of either 1 × 4, 1 × 6, or 1 × 8 (n = 11 patients; 2.3 mm exposed electrode diameter, 10 mm interelectrode distance) or a 2 × 14 [n = 1 patient; 1.2 mm exposed electrode diameter, 4 mm interelectrode distance; high-density (HD) contacts] platinum-iridium contacts (Ad-Tech Medical Instrument). ECoG signals were amplified, online notch filtered (at 60, 120, and 180 Hz, 2nd order Butterworth filter), online bandpass filtered (0.3–250 Hz, 4th order Butterworth filter) and digitized at 1,000 Hz using the Grapevine NIPS. In addition to the down-sampled data, a broadband version of the data was also recorded at 30 kHz (low-pass filtered with a 3rd

order Butterworth anti-aliasing filter at 7500 Hz), which was used for analyses of 300 Hz high-pass filtered high-frequency activity.

## Data pre-processing

All analyses apart from the spike sorting were done in MATLAB (v. 2014b, The MathWorks Inc, Natick, Massachusetts, RRID:SCR_001622). Trials containing artefacts in the ECoG signal were removed after visual inspection, and only recordings with at least eight contralateral grip trials were included. This resulted in 28 recordings (12 single-unit recordings + 16 multi-unit recording) from 12 patients and an average number of 19 ± (SD) 8 contralateral and 18 ± (SD) nine ipsilateral grip trials. The number of units included per patient ranged between 1 and 5 (n = 5: one unit, n = 2: two units, n = 3: three units, n = 2: five units). If subsets of recordings were analysed, then the numbers on the relative contributions are reported in the figure caption.

## Spike sorting

The spike sorting procedure was described in detail elsewhere (*Lipski et al., 2017a*). Single- and multi-unit action potentials were identified offline using principal component analysis (Plexon, Dallas, TX). A cluster qualified as a single unit (SU) if: (1) the principal component cluster was clearly separated from other clusters associated with background activity and other units, (2) if it contained spike waveforms with a unimodal distribution in principal component space, and (3) if the inter-spike interval distribution showed a refractory period of at least 3 ms (*Starr et al., 2003*; *Schrock et al., 2009*). For some SU recordings, the location of the principal component cluster drifted gradually, presumably due to a shift in distance between the electrode and the neuron caused by movement of the brain. Other recordings were classified as multi-unit (MU) recordings. In these, the principle components cluster seemed to include waveforms from multiple units, forming multimodal principal component distributions that could not be clearly separated, or that failed to meet the criterion of having a refractory period of 3 ms.

## Quantification and statistical analysis

### Time-frequency decomposition

Time-varying power and phase were obtained by band-pass filtering the data using Butterworth filters (4$^{th}$ order, two-pass, using fieldtrip functions *ft_preproc_lowpassfilter* and *ft_preproc_highpassfilter Oostenveld et al., 2011*) and calculating the Hilbert transform. The time-frequency plots include the following frequency bins: 5–8 Hz for theta, 8–12 Hz for alpha, 12–20 Hz for low beta, 20–30 Hz for high beta followed by centre frequencies ranging from 40 to 150 Hz with a bin width of 20 Hz (incrementing by 10 Hz).

### Event-related analyses of phase-coupling

Phase locking values (PLV) can be obtained by calculating the length of the average of vectors, with each vector having a length of one on a unit circle and an angle corresponding to the ECoG phase ($\phi$) coinciding with each STN spike (*Lachaux et al., 2000*). It was calculated as follows (n = number of spikes, $\phi_t$ = ECoG phase at the time of one spike):

$$\text{PLV} = \left| \frac{\sum_{t=1}^{n} e^{i*(\phi_t)}}{n} \right|$$

PLVs are bounded between 0 and 1, indicating zero or perfect phase coupling respectively. One important issue when calculating PLVs is that they are inflated when sample sizes are small. To correct for differences in sample size, the pairwise phase consistency (PPC) can be used instead as an unbiased estimator when a sufficiently large (>50) sample is available (*Vinck et al., 2010*; *Aydore et al., 2013*):

$$PPC = \frac{n}{n-1} \left( PLV^2 - \frac{1}{n} \right)$$

The PPC can attain negative values, because in the absence of phase locking, the expected PPC value for infinite amounts of data is zero, and the actual PPC values for limited amounts of data

distribute around zero. Recordings with negative PPC values were excluded for analyses investigating phase offsets between contra- and ipsilateral grip trials as described below.

The fact that the firing rate of some cells increased or decreased around movement or cue onsets has important implications: If we would choose a fixed window size for calculating the PLV or PPC, we would base our analysis on variable and sometimes low numbers of spikes. Variable spike numbers would results in variable biases of the PLV metric, and low spike numbers in noisy estimates of the PPC metric. Another challenge was that some patients had substantially slower reaction times or longer grip durations than others because in the intraoperative setting, task constraints were not very strict.

To get comparable estimates of the coupling strength around each of the key trial events, we implemented the following custom-written event-locked variable-window width PLV estimation procedure: First, each interval between neighbouring task events was subdivided into eight equidistant points (including the events) resulting in 29 time points for each trial (because the last point of an interval was identical to the first point of the next interval the points were concatenated the following way: [1-8, 2-8, 2-8, 2-8]). We had four intervals in total (Right/Left Cue to Go Cue, Go Cue to Grip Onset, Grip Onset to Grip Offset, Grip Offset until the end of a 0.4 s long post-movement window). The first time point of each trial was set at the onset of the Right/Left Cue and the final one at 0.4 s after the grip offset. This procedure ensures that a given plot shows unbiased estimates of the coupling strength around all key task events and not just for one single event, which would be the case if it was locked only to movement or cue onsets.

To include the same number of spikes around each time point, and thus avoid fluctuations of the PLV just because of differences in sample size, the windows centred around each time point were scaled in width. Separately for each recording, the target spike number was set to the average number of spikes occurring within a 0.3 s window (obtained by summing up all spikes across all trials, dividing by the total trial length and multiplying by 0.3 s). If this number was below 50, it was set to 50 to avoid very small and thus less representative samples. Each window was placed symmetrically around each time point and its width was set such that the sum of spikes across all trials would match the target number as closely as possible (*Figure 3—figure supplement 1*). This procedure allowed an unbiased comparison of coupling over time despite changes in firing rate that occurred in some recordings. The average window size was $0.32 \pm$ (SD) 0.04 s (min.: 0.16 s, max.: 0.75 s). Note that the procedure did not ensure that the number of spikes was the same across subjects. Ensuring this would have required an even larger variability of window widths. The average number of spikes per window was $171 \pm$ (SD) 121 spikes. However, our main results hold both for PLV and PPC values, with the latter eliminating biases resulting from different sample sizes.

Each matrix was composed of 16 frequency bins and 29 time points. The resulting 28 matrices (one for each of the 28 recordings) were event-locked to all four task events and could thus be easily averaged across recordings despite differences in reaction times or grip duration. The coarsely sampled matrices were smoothed using the MATLAB function *interp2* with an interpolation factor of 2 resulting in a grid with a resolution of 61 * 113. Finally, the time-dimension was resized using the MATLAB function *imresize* to rescale the inter-event intervals to the average recorded durations.

## Firing characteristics

Firing rates were calculated within a 300 ms window around each time point to get a time-evolving estimate. Three additional firing characteristics were calculated similar as in *Rule et al. (2017)*: The interspike interval coefficient of variation (ISI CV), the percentage of bursting (%burst) and the mode of the ISI (ISI mode). Because properties such as the coefficient of variation (CV) can again be biased by small sample sizes and to enable a direct comparison, we used the same window widths to calculate time-evolving firing characteristics as outlined above for calculating the time-evolving PLV. A robust version of the ISI CV was calculated as the median absolute deviation of all ISIs divided by the median and multiplied by 100. The %burst metric was defined as the percentage of all ISIs shorter than 10 ms. To calculate the ISI mode, we first excluded all bursts, and then computed a probability density estimate (using the MATLAB function *ksdensity*) of which the mode was extracted. The baseline firing characteristics that were used to test for significant deviations were computed by splitting the whole recording into segments with the same number of spikes as were included when computing it across trials or into 300 ms long segments for computing the baseline

firing rates. To get a robust estimate of individual baseline values, the median of all segments was computed for each individual recording. The 28 recordings were also assessed individually with cluster-based permutation tests as shown for the peri-stimulus time histograms in *Figure 3—figure supplement 3*. Cells were classified as increasing or decreasing their firing rates if a significant cluster occurred between the Go cue and the movement offset.

## ECoG contact position normalization to MNI space

The temporarily placed subdural ECoG strips were localized by aligning intraoperative fluoroscopy (x-ray) images of the strip to co-registered preoperative MRI and postoperative CT scans (*Randazzo et al., 2016*). In brief, preoperative MRI scans were co-registered to postoperative CT using SPM (http://www.fil.ion.ucl.ac.uk/spm/software/spm12/) and consecutively decomposed as 3D skull and brain surfaces using Freesurfer (*Dale et al., 1999*; https://surfer.nmr.mgh.harvard.edu/). The reconstructions were then co-registered to common landmarks visible in the intraoperative fluoroscopy images (stereotactic frame pins, implanted DBS electrodes and skull outline). The parallax effect of the fluoroscopic images was accounted for by using the measured distance from the radiation source to the subject's skull to adjust the projection of the skull/brain surfaces. Three-dimensional coordinates of ECoG recording sites were obtained in native Freesurfer space, based on the alignment of the ECoG strip in the fluoroscopic image to the cortical surface reconstruction. To allow group analysis, the electrode locations were then transformed to ICBM152 population averaged cortex space using population-based normalization of the native surface reconstructions (*Saad and Reynolds, 2012*).

## ECoG contact selection

To get a local estimate of the cortical power and phase, we computed bipolar configurations by subtracting neighbouring ECoG contacts resulting in n-1 bipolar channels if n contacts were recorded. For each spike recording, the ECoG contact with the highest PLV between 60–80 Hz and within −0.1–0.4 s around movement onset during contralateral gripping was pre-selected for further analyses. 60–80 Hz was chosen, as previous research has shown coherence between STN LFP and cortical oscillations in this frequency band (*Williams et al., 2002*; *Litvak et al., 2012*). We also performed the same selection procedure to find a separate set of ECoG contacts showing the highest coupling during ipsilateral gripping, which was used to perform control analyses. *Figure 2—figure supplement 1* shows the topographic distribution of coupling strength in one example recording during contra- and ipsilateral gripping.

## Topography of gamma coupling and power

In the topoplots of movement-related gamma and beta power changes, the average across all contacts was displayed. Relative power from a −0.1–0.4 s window around grip onsets was normalized by the average power from the same contact across the whole recording. All points of the cortical mesh within a 4 mm radius (to increase spatial specificity when including all contacts) from the corresponding ECoG site were coloured uniformly with the corresponding value and then averaged.

To illustrate the cortical distribution of gamma coupling during contra- and ipsilateral gripping, we displayed the two sets of ECoG sites with highest gamma coupling during contralateral gripping as well as during ipsilateral gripping. We also show the distribution of the subset of contacts displaying significant movement-related spike-to-60–80 Hz gamma coupling in the same window. Significance was assessed again by creating permutation distributions. To pass the test, two conditions had to be met: First, the original PPC should be significantly higher than the null-distribution created by randomly permuting the ECoG LFP-to-STN spike association 500 times across trials. With this method, the pattern of spikes and the more pronounced gamma synchrony in the ECoG LFP at movement onset remains intact. Second, the original PPC should be significantly higher around movement onset compared to a pre-movement period (spikes were drawn from a window ranging from −3 to −2 s before movement onset and matched in the number of spikes) from where spikes and coincident ECoG phases were extracted. We applied one-sided tests ($\alpha = 0.1$, but both conditions had to be significant) such that a larger number of contacts, which is more representative of the overall distribution, was displayed in the topoplot. Cortical mesh areas within a 7 mm radius

around each bipolar contact site (defined as the average location between the two contacts used for the bipolar calculation) were displayed as significant to ensure good visibility of each single contact.

## Cluster-based permutation procedure

Significance tests in plots showing multiple time- or frequency points were performed using a cluster-based permutation procedure for multiple-comparison correction (*Maris and Oostenveld, 2007*). Condition labels of the original samples (e.g. median-split fast or slow RTs) were randomly permuted 1000 times such that for each recording the order of subtraction can change. When firing rates in individual recordings were compared relative to a baseline period (ranging from −0.5 s before the Left/Right cue to the Go signal), the normalized firing rate was compared against 0 by flipping the sign of the firing rates of a subset of randomly chosen trials. This permutation procedure results in a null-hypothesis distribution of 1000 differences. Suprathreshold-clusters (pre-cluster threshold: $p<0.05$) were obtained for the original unpermuted data and for each permutation sample by computing the z-scores relative to the permutation distribution. If the absolute sum of the z-scores within the original suprathreshold-clusters exceeded the 95[th] percentile of the 1000 largest absolute sums of z-scores from the permutation distribution, it was considered statistically significant.

To test if PLVs were significantly elevated, we created a permutation distribution by shuffling the LFP-to-spike relationship in each recording 500 times. The same windows were chosen as for the calculation of the original PLVs but the trial-association between the ECoG LFP and STN spikes was permuted such that, for example, the spike train from trial one was paired with the LFP from trial 4, resulting in phases that could have been observed by chance if no spike-to-phase coupling existed. Importantly, the same shuffling was applied to all time- and frequency points within each of the 500 permutations. If different randomizations were applied for each time- or frequency point, the natural appearance of clusters would be prevented, and the test, which is again based on the z-scores of the largest suprathreshold clusters as above, would not be sufficiently conservative.

Whenever time-frequency plots were compared between contra- and ipsilateral grip trials, the cluster-based permutation procedure was performed on the differences (e.g. between PLVs during fast vs. slow RTs or between original PLVs and shuffled PLVs).

## Correlation between coupling strength following the Go cue and RTs

To test for a relationship between coupling strength and RTs, Pearson's correlation coefficients were computed for each recording as follows: Trials were sorted according to their RTs. All spikes within a 0.5 s window just after the Go cue were ordered accordingly, and their corresponding cortical gamma phases were determined. The resulting array of spike-LFP gamma phases was then subdivided into seven non-overlapping bins that contained an equal number of spikes to compute one PLV for each bin. The average RT for each bin was obtained by taking the mean of the RTs corresponding to each of the included spikes. Correlation coefficients were computed based on these seven bins, Fisher's Z-transformed and subjected to a t-test to assess if they significantly differed from zero on the group level.

## Phase-dependent spiking probability and polarity flipping according to local high-frequency activity

Average PLVs can be high across recordings although the average preferred phase could differ from recording to recording. When computing the PLV, the information of preferred phase is not retained as only the vector length of the average phase vector is taken into account, reflecting how strongly bundled or spread the phase values are. To test whether spikes were consistently more probable at certain phases and less probable at others, we computed the spiking probability in four non-overlapping phase bins across the evolving gamma cycles. An issue that needs to be dealt with beforehand is that the phase of the cortical oscillation can be polarity-reversed depending on the order of subtraction of the two channels for the bipolar configuration and the location and orientation of the gamma-generating source. If the order of subtraction of two channels would be flipped, then the peaks of the oscillation would turn into the troughs, and the troughs into peaks. Group statistics of spiking probabilities across all recordings can thus only be meaningfully computed after standardizing the polarity. To perform an automated polarity-standardization procedure, we took into

account that gamma oscillations can capture fluctuations in local excitability and computed the local high-frequency activity (HFA) as a proxy of background unit activity. This was performed on the ECoG data sampled at 30 kHz, which was recorded without any online filters. The HFA was computed by high-pass filtering the ECoG signal at 300 Hz, full-wave rectifying and low-pass filtering with a cut-off of 100 Hz. This is a commonly used procedure (e.g. *Eckhorn et al., 1988*; *Pooresmaeili et al., 2010*) with the cutoff of 300 Hz considered a good threshold to obtain an estimate of multi-unit activity (*Logothetis, 2003*) as the energy of spike waveforms is in that frequency range, although its precise value is not critical. Using a cutoff of 300 Hz made sure that we could detect fast fluctuations of activity. This would have been more difficult if a cutoff in the vicinity of 60–80 Hz were used because the signal would then likely be dominated by slower fluctuations in lower frequencies. A 150 Hz high-pass filter, or alternatively computing the amplitude of the analytic signal instead of full-wave rectification, provided the same results. The low-pass filtering results in a smoother version of the >300 Hz rectified signal, reducing fast fluctuations and resulting in a smoother co-modulation of the HFA with the peaks and troughs of the evolving 60–80 Hz gamma signal. Importantly, increases in background unit activity close to the recording sites, i.e. detecting when > 300 Hz amplitude is increased, are invariant to the order of subtraction of the two neighbouring ECoG contacts that is performed to initially compute the bipolar signal (see *Figure 5—figure supplement 1*). Hence the HFA is an ideal feature to standardize the polarity of the bipolar signal across all recordings. Next, we extracted the 60–80 Hz gamma phase from the same signal (which was offline notch-filtered) and subdivided it into 126 equally spaced, non-overlapping phase bins with a bin width of $0.05\pi$. For each bin, the average HFA was computed. The resulting 126 points-long vector was smoothed with a moving average filter (using the MATLAB function *smooth*, width = 20 samples). To increase the signal-to-noise ratio, the whole recording was used to compute how the HFA co-fluctuated with the gamma phase. To check if the polarity of the gamma signal should be flipped, the average HFA was calculated within a $0.4\pi$ wide window centred at the middle (at $0\pi$ of the gamma phase, which is where the gamma-filtered data would have its peak) and an average of the two pieces on the side ($0.2\pi$ wide on each side, where the gamma-filtered data would have its minima). If the average HFA from the middle was lower than the average HFA from the sides, the bipolar signal was flipped (see *Figure 5—figure supplement 2*). All cortical recordings showed a distinguishable peak, and thus this method helped to ensure that the cortical gamma signal was flipped such that the gamma peak consistently coincided with increases in the HFA.

Finally, to compare changes in spiking probability across the gamma cycle, we split the gamma phase into four non-overlapping phase-bins resulting in bins with a width of $0.5\pi$. Four bins were chosen to allow for a resolution that allowed us to detect differences without resulting in too many multiple comparisons, which needed to be corrected for. The movement-related spiking probabilities for the four phase bins were computed based on data from a −0.1 to –0.4 s window around movement onset. To ensure that the effects are movement-onset-specific, these probabilities were normalized by baseline spiking probabilities, which were computed within a same sized window starting 3 s before movement onset. The relative changes were obtained by subtracting baseline spiking probabilities from movement-related spiking probabilities.

Two-factorial 4 (bin) x 2 (effector side) repeated-measures ANOVAs were computed in SPSS (IBM Statistics SPSS 22, RRID:SCR_002865), to compare spiking probabilities across the four phase-bins. Q-Q plots of the residuals were visually inspected to exclude strong deviations from normality. If the sphericity assumption was violated, Huynh-Feldt correction was applied. As additional analysis, the same two-factorial ANOVA was also computed with five instead of four phase bins. Subsequent pairwise comparisons were performed using t-tests or Wilcoxon signed-rank tests if the normality assumption (assessed by Lilliefors tests) was violated.

## Comparison of phase offsets between contra- and ipsilateral gripping

Another key question was whether the timing of STN spikes relative to the cortical gamma phase (from the ECoG sites that showed the strongest coupling during contralateral gripping) systematically differed between contra- and ipsilateral grip trials. Note that because the polarity of the gamma signal was standardized based on the HFA from the whole recording, the polarity was the same for contra- and ipsilateral grip trials. Because here we directly compare the phase difference between contra- and ipsilateral gripping, a difference of 1pi, for example, would remain the same

even if the polarity of the whole signal would be flipped, as may occur with bipolar derivations. We computed the circular median of all phases coinciding with STN spikes around movement onset (in a −0.1–0.4 s window). Only recordings with positive PPC values were included resulting in n = 23 as an estimate of the average phase would not be meaningful if the phases were relatively uniformly distributed. The angle difference between contra- and ipsilateral gripping was first tested against zero by computing confidence intervals based on the 23 angle differences (*circ_confmean* function from the *circ_stats* toolbox *Berens, 2009*). We decided not to scale the weight of the angles by the number of spikes included or the PPC strength for each recording as this would bias the average across all recordings towards few data points. Spike numbers were not subsampled, which could have been a step to match spike numbers between recordings, as the estimate of the mean angle only gets more precise with a larger number of representative samples. We also investigated whether the offset of 180˚ degrees, which became apparent, was specific in time by plotting the time course of the p-values of a V-test (*circ_vtest* function from the *circ_stats* toolbox *Berens, 2009*). A V-test examines the phase differences for circular uniformity, similarly as a Rayleigh test, but is more powerful because a mean difference – in this case 180˚ – can be specified. Note that the average preferred phase, as compared here, is independent of the coupling strength used to select the ECoG sites. Detecting a significant effect would thus be unrelated to the ECoG selection procedure, however because coupling was strongest around movement onset, the effect would be expected to be temporally specific to windows around movement onset.

## Acknowledgements

P Fischer and PB were funded by the Medical Research Council (MC_UU_12024/1). R M R was supported in part by the Walter L Copeland Fund of The Pittsburgh Foundation. R S T was supported by the NIH (R01 NS091853-01A1). R S T and R M R received further NIH support (R01 NS110424-01 CRCNS). W J L was supported by National Institute of Mental Health Grant R01MH107797 (PI, Avniel Ghuman; co-investigator, R M R). P Fries acknowledges grant support by DFG (SPP 1665, FOR 1847, FR2557/5-1-CORNET, FR2557/6-1-NeuroTMR), EU (HEALTH-F2-2008-200728-BrainSynch, FP7-604102-HBP, FP7-600730-Magnetrodes), a European Young Investigator Award, NIH (1U54MH091657-WU-Minn-Consortium-HCP), and LOEWE (NeFF).

## Additional information

### Funding

| Funder | Grant reference number | Author |
|---|---|---|
| Medical Research Council | MC_UU_12024/1 | Petra Fischer<br>Peter Brown |
| National Institute for Health Research | R01 NS091853-01A1 | Robert S Turner |
| National Institute for Health Research | R01 NS110424-01 CRCNS | Robert S Turner<br>Robert Mark Richardson |
| National Institute of Mental Health | R01MH107797 | Witold J Lipski<br>Robert Mark Richardson |
| Deutsche Forschungsgemeinschaft | SPP 1665 | Pascal Fries |
| Deutsche Forschungsgemeinschaft | FOR 1847 | Pascal Fries |
| Deutsche Forschungsgemeinschaft | FR2557/5-1-CORNET | Petra Fischer |
| Deutsche Forschungsgemeinschaft | FR2557/6-1-NeuroTMR | Petra Fischer |
| National Institute for Health Research | 1U54MH091657-WU-Minn-Consortium-HCP | Pascal Fries |
| LOEWE Zentrum AdRIA | NeFF | Pascal Fries |

| Pittsburgh Foundation | Walter L Copeland Fund | Robert Mark Richardson |
| --- | --- | --- |
| European Union 7th Framework Programme | HEALTH-F2-2008-200728-BrainSync | Pascal Fries |
| European Union 7th Framework Programme | FP7-604102-HB | Pascal Fries |
| European Union 7th Framework Programme | FP7-600730-Magnetrode | Pascal Fries |

The funders had no role in study design, data collection and interpretation, or the decision to submit the work for publication.

## Author contributions
Petra Fischer, Conceptualization, Data curation, Software, Formal analysis, Validation, Visualization, Methodology, Project administration; Witold J Lipski, Conceptualization, Resources, Data curation, Investigation, Methodology, Project administration; Wolf-Julian Neumann, Resources, Methodology; Robert S Turner, Methodology; Pascal Fries, Supervision, Methodology; Peter Brown, Supervision, Funding acquisition, Methodology; R Mark Richardson, Conceptualization, Resources, Data curation, Supervision, Funding acquisition, Investigation, Methodology, Project administration

## Author ORCIDs
Petra Fischer  https://orcid.org/0000-0001-5585-8977
Witold J Lipski  http://orcid.org/0000-0003-1499-6569
Wolf-Julian Neumann  https://orcid.org/0000-0002-6758-9708
Robert S Turner  http://orcid.org/0000-0002-6074-4365
Pascal Fries  http://orcid.org/0000-0002-4270-1468
Peter Brown  http://orcid.org/0000-0002-5201-3044
R Mark Richardson  https://orcid.org/0000-0003-2620-7387

## Ethics
Human subjects: Patients provided written, informed consent in accordance with a protocol approved by the Institutional Review Board of the University of Pittsburgh (IRB Protocol no. PRO13110420).

## Decision letter and Author response
Decision letter https://doi.org/10.7554/eLife.51956.sa1
Author response https://doi.org/10.7554/eLife.51956.sa2

# Additional files

## Supplementary files
• Source code 1. Code to generate the time-frequency figures in the main manuscript and in the supplementary figures including the functions to run the cluster-based permutation statistics.

• Transparent reporting form

## Data availability
We have provided the data and the code (including the functions to run the cluster-based permutation statistics) with which one can generate the time-frequency figures in the main manuscript and in the supplementary figures (Fig. 3, 4, Fig. 3- figure supplement 5 and Fig. 4-figure supplement 2).

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
