## [Decision Letter]

**Acceptance summary:**

Despite the importance of movement for nearly all behaviors, questions still remain about how different brain areas work together to coordinate physiological movements. Here the authors report findings from a unique dataset recorded from Parkinson's disease patients undergoing neurosurgery. Their results characterize interactions between motor cortex electrocorticography and subthalamic nucleus unit activity in humans during performance of a motor task. Their findings provide insight into coordination between subcortical and cortical notes during task-based behavior.

**Decision letter after peer review:**

Thank you for submitting your article "Movement-related coupling of subthalamic nucleus spikes to cortical γ" for consideration by *eLife*. Your article has been reviewed by three peer reviewers, including Nicole C Swann as the Reviewing Editor and Reviewer #1, and the evaluation has been overseen by Laura Colgin as the Senior Editor.

The reviewers have discussed the reviews with one another and the Reviewing Editor has drafted this decision to help you prepare a revised submission.

Summary:

In this manuscript the authors recorded intraoperative electrophysiological data from 12 Parkinson's disease patients during a movement task. Electrocorticography and concurrent STN unit recordings were obtained. The authors conducted a number of analyses to describe the electrophysiological patterns during the movements, with special interest in examining interactions between STN units and motor cortex field potentials and comparing patterns between hemispheres (i.e. contra-lateral compared to ipsi-lateral activity, relative to the hand moved). A core finding is that prior to movement there was elevated STN firing to motor cortical field potential locking in the γ (>30 Hz) range. This was more pronounced for contralateral compared to ipsilateral interactions, and not explained by higher STN firing. Next the authors showed that these interactions were modulated as a function of motor speed. Specifically, the phase locking between units and γ ECoG was higher for fast compared to slow movement. The authors further probed these relationships using other analyses focusing on γ phase with respect to unit firing.

Overall, the reviewers found this manuscript to be methodologically impressive and appreciated the challenge of obtaining intraoperative unit and ECoG recordings in humans during a task. The overall finding of increased interaction between sub-cortex and cortex as a function of movement (and movement speed) is also interesting. However, we all had difficulty understanding some of the analyses, particularly the phase-related analyses presented in Figures 5 and 6. Our confusion was not necessarily about what was done, but why and what the biological meaning or significance would be for the results from these analyses. We were concerned this might limit the overall impact of this manuscript for a wider audience. As such, this was the biggest barrier to our enthusiasm for this manuscript, but given our otherwise positive evaluation, we elected to invite a revision. We elaborate on this point and others below.

Essential revisions:

1) A major barrier for this manuscript, in terms of appeal to a wider audience, is the complexity of the analyses used. A challenge to this kind of work, especially using novel approaches like ECoG and STN together, is that a vast search space is opened with multiple electrodes, frequency ranges, time points, units etc. Although the initial analyses (Figures 1-3) were clearly justified, some of the subsequent analysis (such as those associated with Figures 5 and 6) felt a bit post-hoc. It was not always clear why certain approaches were being used and what we learned about the brain, motor system, PD, etc. from the analyses presented. We believe the manuscript would benefit from an easier to follow rationale for each analyses, presented upfront, and clear delineation of which analyses were planned (and why) and which were exploratory (and what statistical corrections were applied).

2) In the manuscript there is not a clear distinction between 'oscillatory' γ (or narrow range γ) and broadband γ. Although both are (unfortunately) called 'γ' in the literature, the emerging view is that the two have very different etiologies with broadband γ being a surrogate for neural firing and the narrow γ being a 'true' oscillation. In places, the authors seem to have made this distinction, specifying narrow γ and showing figures where γ is focused in a relatively narrow range (Figure 2A). However, in other portions of the manuscript the distinction is unclear. For instance, in Figure 2B there is clearly a broadband γ response which is expected to occur in motor cortex during movement (see Crone, 1998, for instance), but it is presented with a narrowband result (Figure 2A)? Do the authors believe these two examples reflect the same underlying process? Can more explanation be given please?

We appreciate it may not always be possible to differentiate whether a particular result is driven by broadband or oscillatory γ but it would be helpful for the authors to spend some time clarifying that they are different proposed etiologies for γ activity in the Introduction and interpret their results with this context. The authors should also show that there is indeed a γ oscillation present in the signal, by showing a power spectral density plot, for example. This is important because if the effects are primarily driven by broadband γ, then it makes less sense to extract phase (as there would be no oscillation present).

3) There were 28 units from 12 people so there is some dependency with the analyses. Including independent and dependent samples in the same analyses can be problematic (see Aarts et al., 2014, Nature Neuroscience). The authors should, at least, report how many units each patient contributed and how many units per participant were included in each analysis so that the reader can assess if effects may have been driven by a subset of patients.

4) We all really struggled to understand some of the phase analyses. For instance, we understand the need to invert the phase of a raw signal, but it was not clear to us why a signal reflecting γ phase, derived from γ amplitude, would need to be flipped. We also did not understand the criteria for deciding a flip needed to be performed. Please clarify these analyses.

It was also confusing to even understand what HFA was meant to reflect. It is defined in the beginning of the manuscript as simply "high frequency activity" which might be expected to reflect amplitude of higher frequency activity (which might be derived simply by high pass filtering signal and then computing amplitude from the complex signal) but later very different procedure was described to calculate it, which we did not understand. Please clarify what HFA is meant to reflect and how the reported analysis represents it.

5) In Figure 3. It is interesting that higher coupling (between fast and slow trials) starts so sharply and exactly at the Go signal. Do the author's have an idea why it could be so precise? Could it just appear to be occurring at this time because of window used? The timing makes is seem almost artifactual. Is this somehow a by-product of analysis window selection?

6) Could there be differences in baseline motor function that could account for some of the differences seen between contralateral and ipsilateral gripping? For example, if recordings were always done in more (or less) severely affected hemisphere of the PD patients, this could explain some of the electrophysiologic differences seen between contralateral and ipsilateral gripping. Likewise, dominant hand should also be reported since this might impact the hemisphereic findings.

7) Determining strip location using fluoroscopy would result in a very coarse localization. Was fluoroscopy performed in multiple planes? Since their location-related findings make sense (movement related changes in the precentral gyrus), localization was presumably accurate, but nonetheless, limitations to the localization should be noted in the Limitations section.

8) Please add trial numbers for each participant.

9) Given that narrowband γ activity has been associated with pathophysiology (i.e. involuntary movements), please add information about the relationship between the disease state/scores and the magnitude of the coupling. Is this event-related coupling modulated at all by disease or is it purely a movement-related effect?

10) Figure 2C. Was the firing rate analyzed using the same variable windows described for coupling analysis? Or do they report instantaneous firing rate? Would results be different if alternative method was used?

[Editors' note: further revisions were suggested prior to acceptance, as described below.]

Thank you for resubmitting your article "Movement-related coupling of human subthalamic nucleus spikes to cortical γ" for consideration by *eLife*. Your article has been reviewed and extensively discussed by the original three peer reviewers, including Nicole C Swann as the Reviewing Editor and Reviewer #1, and the evaluation has been overseen by Laura Colgin as the Senior Editor.

Summary:

The authors have addressed many of our comments, but we do have some additional concerns.

First, although some points of the methods have been well-clarified, others are still quite difficult to understand. We elaborate on this below. Additionally, now that we better understand the methods used, this has raised some additional questions which we discuss below. Overall, the results that correspond to Figures 1-4 are clear and well-motivated, but the later ones remain challenging to interpret.

Essential revisions:

– Although improved, many aspects of the methods and results are still very difficult to understand. It is often difficult to understand what data went into what analyses and what each analysis reflects. One option would be to include a schematic figure to illustrate this. Also, in the results, it might be helpful to move some of the details to the Materials and methods and try to explain the analyses more conceptually – i.e. what was the analysis testing [i.e. what do the results mean] and what did you find? For instance, some of the detail in subsection “STN spiking probability is modulated relative to the cortical γ phase” seems more appropriate for methods.

– For the electrode selection method – please add a bit more detail. The authors state that the electrodes with strongest spike-to-γ coupling were used. Strongest compared to what? Strongest per subject? Strongest relative to group? It would be helpful to get a sense of what the other electrodes showed? Was there no coupling? Or did all electrodes show some coupling, but not as much. Perhaps at least an example of coupling for all electrodes in an example subject would help give a better sense of the data.

– Analysis in Figure 5C seems a bit biased. Since electrodes were selected based on strongest coupling in this time-range, is it surprising that the phase is most consistent there as well? Other time periods would be expected to have systematically less coupling in general, right? Perhaps because they have inconsistent phase? This should be corrected or at least, this potential confound should be discussed.

– In the Materials and methods it is mentioned that the ECoG is filtered with a bandpass cutoff of 250 Hz, so how can the HFA include activity at >300 Hz? Further to this point, from the Materials and methods, it sounds as the filtering has been done "online". From the writing it is not clear if the entire filtering (notch filtering and also the band pass filtering) was done online or only the notch filtering was done online. If former is the case, doesn't this mean that the recorded data was filtered at 250 Hz and then stored by the recording system? Then how was it possible to analyze >300 Hz activity if it was above the filtering cut off? Please clarify this point. Also please add filter details for bandpass filter.

– The rationale for calculating HFA is not clear. The authors cite Logothetis, 2003 for the selection of the 300 Hz cutoff, however, it seems this paper used the 300 Hz cutoff for multi-unit activity, and here the authors are using ECoG. STN LFPS focus on higher ranges, but in ECoG, broadband γ as a surrogate for neural firing extend much lower (extending down to 70 Hz), and have generally not been reported >300 Hz. If the broadband signal really does extend this low, there would be overlap with the narrowband signal, which would complicate the coupling analysis. Finally, could the authors clarify how the use of the full wave reactivation compares to taking the analytic amplitude of the signal? It was not clear why rectification was used as the analytic amplitude approach would be more conventional and perhaps reflect more readily the author's goals (if we understand correctly)? In short, this portion of the paper is hard to understand. Please clarify or perhaps remove from the manuscript.

– The Authors mention that they use notch filter to at 60 Hz to remove line noise. Please specify the notch filter characteristics. This is important given that the 60-80 Hz frequency range is used to analyze γ activity which partly includes the line noise frequency range.

– Some of the interpretations in subsection “Cortical high-frequency activity increases faster in sites that show strongest γ coupling during contralateral gripping” seem very definitive when maybe they are really more hypotheses. The authors may consider toning town the language a bit.

– For our original point 9 – The author's explanation makes sense, but the variability of relationships between unit to ECoG coupling within a person should be stated somewhere in the manuscript. The authors may also want to mention that this is why across subject's correlations related to clinical severity cannot be examined.

– Please put the table with patient characteristics and experimental information back in main manuscript (not supplements).

– It would be preferable to include the task schematic and a bit more description in the main paper, not just the supplements.

– For the comparison of phase offset between contra- vs ipsi- gripping, one concern is if the 180 degree difference could be due to the bipolar montage (i.e. signals were flipped for a trivial reason). How do the authors know that this is not the case? (Since they describe in detail the need to flip signals for the HFA analysis). We may be mis-understanding here.

– It would be helpful to provide a bit more context about the Cheyne paper and why it was selected for this comparison. Also, although it is somewhat obvious given the paper was published 11 years ago, it might be helpful to clearly state that this was an independent paper, in different participants. Also, related to this, the authors refer to "detected in MEG studies", but then cite just one paper. Perhaps change to "detected in a MEG study".

---

## [Author Response]

Essential revisions:1) A major barrier for this manuscript, in terms of appeal to a wider audience, is the complexity of the analyses used. A challenge to this kind of work, especially using novel approaches like ECoG and STN together, is that a vast search space is opened with multiple electrodes, frequency ranges, time points, units etc. Although the initial analyses (Figures 1-3) were clearly justified, some of the subsequent analysis (such as those associated with Figures 5 and 6) felt a bit post-hoc. It was not always clear why certain approaches were being used and what we learned about the brain, motor system, PD, etc. from the analyses presented. We believe the manuscript would benefit from an easier to follow rationale for each analyses, presented upfront, and clear delineation of which analyses were planned (and why) and which were exploratory (and what statistical corrections were applied).

Thank you for highlighting this. We have now provided a more detailed rationale in the Introduction:

“Inspired by previous findings that have demonstrated strong correlation between movement kinematics and basal ganglia γ-band activity (Brücke et al., 2012; Fischer et al., 2017; Lofredi et al., 2018), we aim to shed light on the underlying neurophysiological mechanism resulting in cortico-subcortical communication in the γ band. Therefore, we set out to examine if STN spike-to-cortical γ phase coupling can predict the timing or vigour of contralateral action initiation. […]

Oscillations in the motor cortical LFP can capture periods of periodically discharging neurons (Donoghue et al., 1998). We aimed to capture this periodic discharge by using a proxy of multi-unit activity – high frequency activity (HFA, here defined as >300 Hz high-pass filtered activity) to examine if those ECoG sites in which γ was coupled most strongly with STN spikes were distinct in showing a steeper increase of HFA, and thus discharge activity, within each γ cycle. Additionally, to further investigate the organization of a widespread cortical γ power increase as an exploratory side question, we examined whether systematic phase shifts provide evidence of information segregation (Maris et al., 2013, 2016; van Ede et al., 2015) between distinct cortical sites during contra and ipsilateral movement.”

We have also added to the Results:

“If STN spikes that occur at one phase of the cortical γ cycle contribute to boosting cortical γ for contralateral movements, then during ipsilateral movements unintentional boosting of γ could be avoided if spikes would occur at the opposite phase (i.e. 180° apart). The following section will quantify if such a systematic difference in spike timing was present. This analysis was also motivated by the above reported differences in spiking probabilities between contra- and ipsilateral gripping observed at a specific phase of the γ cycle.”

And in the section “Cortical high-frequency activity increases faster in sites that show strongest γ coupling during contralateral gripping”:

“The final two analyses were performed to understand how local ECoG γ activity was distinct between different cortical sites during contra- and ipsilateral movements, and were more exploratory in nature.”

We also acknowledged in the Discussion that the final two analyses were exploratory:

“The sign of the shift flipped between contralateral and ipsilateral grip trials, which could potentially indicate a directional change in information flow (Battaglia et al., 2012; Besserve et al., 2015). However, we would like to acknowledge that this analysis was exploratory and that the phase difference was only significant during ipsilateral gripping, where γ power was relatively weak.[…]The final two exploratory analyses investigating phase offsets between γ oscillations in different ECoG sites and the steepness of the HFA increase would also be more informative if a more consistent grid coverage had been present. We nevertheless reported these findings to motivate future studies of movement-related precentral and postcentral γ activity that could be examined with ECoG recordings alone.”

We did not perform any form of statistical corrections for the final two more exploratory analyses, however, it is now clear from the additional text that they were exploratory.

2) In the manuscript there is not a clear distinction between 'oscillatory' γ (or narrow range γ) and broadband γ. Although both are (unfortunately) called 'γ' in the literature, the emerging view is that the two have very different etiologies with broadband γ being a surrogate for neural firing and the narrow γ being a 'true' oscillation. In places, the authors seem to have made this distinction, specifying narrow γ and showing figures where γ is focused in a relatively narrow range (Figure 2A). However, in other portions of the manuscript the distinction is unclear. For instance, in Figure 2B there is clearly a broadband γ response which is expected to occur in motor cortex during movement (see Crone, 1998, for instance), but it is presented with a narrowband result (Figure 2A)? Do the authors believe these two examples reflect the same underlying process? Can more explanation be given please?We appreciate it may not always be possible to differentiate whether a particular result is driven by broadband or oscillatory γ but it would be helpful for the authors to spend some time clarifying that they are different proposed etiologies for γ activity in the Introduction and interpret their results with this context. The authors should also show that there is indeed a γ oscillation present in the signal, by showing a power spectral density plot, for example. This is important because if the effects are primarily driven by broadband γ, then it makes less sense to extract phase (as there would be no oscillation present).

We agree that the distinction between true oscillatory narrow-band and non-oscillatory broadband γ is a difficult issue with ECoG recordings. Broadband γ reflects the shapes of short neuronal events (Ray and Maunsell, 2011), including postsynaptic potentials (PSPs) and action potentials (APs). Each individual PSP has a roughly exponential shape in time, which corresponds to a 1/f shape in frequency (Miller et al., 2011, PLOS Comp Biology), and each AP corresponds to a broad band between 150 and 5000 Hz. At the level of spikes, even closely neighbouring neurons show only weak pairwise correlations (Schneidman et al., 2006, Nature), hence neurons distributed between the STN and cortex are expected to have very weak pairwise spike correlations. Thus, spikes from an individual STN neuron are not expected to show significant locking to broadband cortical γ that captures superimposed APs and PSPs. However, as further discussed below, we think that 60-80 Hz activity captures true oscillations which explains the difference between Figure 2B and 2A.

We also computed the power spectrum based on the -0.1:0.4s window around movement onset, and it looks like there is a distinct peak between 60- 80 Hz (red dashed lines = 60 and 80 Hz). However, the graph may be misleading because a 60 Hz online notch filter was applied during the recordings. Because of this difficulty in interpreting the spectrum and because a distinct peak in the power spectrum is not necessarily a prerequisite for true synchronization (Brunet, 2014), we decided not to include this figure in the manuscript.

**Author response image 1. respfig1:** Power spectrum based on the -0.1:0.4s window around movement onset.

However, we have now added to the Results:

“Movement-related cortical power changes showed a broad power increase between 60-150 Hz (Figure 2B) that is typically observed in ECoG recordings. Such broad power increases can reflect both true oscillatory components as well as non-oscillatory broadband activity capturing short neural events (Ray and Maunsell, 2011). To focus on the 60-80 Hz oscillatory component of interest, we pre-selected the ECoG contacts with the highest PLV between 60-80 Hz over a -0.1 – 0.4s period around movement onset (Figure 1E).”

We have also added to the Discussion:

“Because ECoG signals tend to show a very broad movement-related power increase between 50-200 Hz, past research has primarily focussed on the coupling of spikes to the amplitude of 50-200 Hz activity (Shimamoto et al., 2013; Lipski et al., 2017), which captures aspects of neural activity that are separate from the narrow-band γ phase. […] Our choice of extracting the 60-80 Hz phase also appears valid when considering the clear co-modulation of the amplitude of high frequency activity with the 60-80 Hz filtered signal.”

3) There were 28 units from 12 people so there is some dependency with the analyses. Including independent and dependent samples in the same analyses can be problematic (see Aarts et al. 2014, Nature Neuroscience). The authors should, at least, report how many units each patient contributed and how many units per participant were included in each analysis so that the reader can assess if effects may have been driven by a subset of patients.

“The number of units included per patient ranged between 1 and 5 (n = 5: 1 unit, n = 2: 2 units, n = 3: 3 units, n = 2: 5 units). If subsets of recordings were analysed, then the numbers on the relative contributions are reported in the figure caption.”

We have also added the corresponding numbers for the sub-selection of units to the figure caption of Figure 5 and 6.

4) We all really struggled to understand some of the phase analyses. For instance, we understand the need to invert the phase of a raw signal, but it was not clear to us why a signal reflecting γ phase, derived from γ amplitude, would need to be flipped. We also did not understand the criteria for deciding a flip needed to be performed. Please clarify these analyses.

Thank you for pointing out that this was not clear. Group statistics on how the spike probability changes relative to the γ cycle, for example, and also on the evolving HFA, can only be meaningfully computed if the polarity of the signal that is used as reference for when the γ peaks and troughs occurred is standardized.

We now provide a more detailed explanation with a new supplementary figure and have replaced the potentially confusing term “phase flipping” with “polarity flipping”.

We have added to the Results section:

“The binning was preceded by a procedure to standardize the polarity of the bipolar signals across all recordings (see Materials and methods, Figure 4—figure supplement 1 and 2). Group statistics across all patients can only be meaningfully computed after flipping each cortical bipolar signal according to whether the peak of the γ-filtered bipolar signal coincides with the peak of the co-modulated HFA or with its trough. Both are equally likely and depend on the order of subtraction of the two ECoG signals that were used to derive the spatially focal bipolar signal (see Figure 4—figure supplement 1). Hence the polarity needs to be standardized.”

It was also confusing to even understand what HFA was meant to reflect. It is defined in the beginning of the manuscript as simply "high frequency activity" which might be expected to reflect amplitude of higher frequency activity (which might be derived simply by high pass filtering signal and then computing amplitude from the complex signal) but later very different procedure was described to calculate it, which we did not understand. Please clarify what HFA is meant to reflect and how the reported analysis represents it.

When introducing the term HFA, we have now added a definition:

“We aimed to capture this periodic discharge by using a proxy of multi-unit activity – high frequency activity (HFA, here defined as >300 Hz high-pass filtered and full-wave rectified activity) […]”.

And to our description (“First, the local high-frequency activity (HFA) was computed by high-pass filtering the ECoG signal at 300 Hz, full-wave rectifying and low-pass filtering at 100 Hz.”):

The low-pass filtering is part of a classical procedure to obtain a measure of MUA (see for example Eckhorn et al., 1988) and reduces the fast fluctuations of the >300Hz amplitude to obtain a smoother average, as you can see in Author response image 2:

**Author response image 2. respfig2:** Low-pass filter effect.

We have now expanded this section (without adding the figure):

“First, the local high-frequency activity (HFA) was computed by high-pass filtering the ECoG signal at 300 Hz, full-wave rectifying and low-pass filtering with a cut-off of 100 Hz. This is a commonly used procedure (e.g. Eckhorn et al., 1988; Pooresmaeili et al., 2010)with the cutoff of 300 Hz considered a good threshold to obtain an estimate of multi-unit activity(Logothetis, 2003), although its precise value is not critical. The low-pass filtering results in a smoother version of the >300 Hz rectified signal, reducing fast fluctuations and resulting in a smoother co-modulation of the HFA with the peaks and troughs of the evolving 60-80 Hz γ signal. Importantly, detecting multi-unit activity close to the recording sites, i.e. detecting when >300 Hz amplitude is increased, is invariant to the order of subtraction of the two neighbouring ECoG contacts that is performed to initially compute the bipolar signal (see Figure 4—figure supplement 1). Hence the HFA is an ideal feature to standardize the polarity of the bipolar signal across all recordings.”

5) In Figure 3. It is interesting that higher coupling (between fast and slow trials) starts so sharply and exactly at the Go signal. Do the author's have an idea why it could be so precise? Could it just appear to be occurring at this time because of window used? The timing makes is seem almost artifactual. Is this somehow a by-product of analysis window selection?

The windows centered around the GO signal to compute the coupling strength spanned on average -0.16:0.16s, hence the difference did not necessarily start as sharply as it may look like in the figure. Moreover, the green colour extends slightly to the left of the Go signal. The onset of the effect looks very sharp only because the outline of the significant cluster starts just at the dashed line. The difference cannot be a by-product of the window selection procedure, as the -0.1:0.4s window around grip onset used for the selection procedure occurs much later.

6) Could there be differences in baseline motor function that could account for some of the differences seen between contralateral and ipsilateral gripping? For example, if recordings were always done in more (or less) severely affected hemisphere of the PD patients, this could explain some of the electrophysiologic differences seen between contralateral and ipsilateral gripping. Likewise, dominant hand should also be reported since this might impact the hemisphereic findings.

We have now added information on the recorded hemispheres, handedness, more severe side of motor symptoms and the number of trials for each patient. We also compared if symptom severity was worse on the contra- or ipsilateral side and now report in the article that symptoms tended to be worse ipsilateral to the recorded STN/motor cortex, which may also contribute to the difference seen between contra- and ipsilateral gripping. That the symptoms were worse on the ipsilateral side was merely a coincidence and not a planned part of the protocol.

7) Determining strip location using fluoroscopy would result in a very coarse localization. Was fluoroscopy performed in multiple planes? Since their location-related findings make sense (movement related changes in the precentral gyrus), localization was presumably accurate, but nonetheless, limitations to the localization should be noted in the Limitations section.

Thank you for pointing out the lack of clarity in the description of the localization procedure. All electrode locations were obtained using a published and CT validated approach by projecting the intraoperative fluoroscopy to common fiducials/landmarks on the volumetric pre and postoperative images to determine the closest brain surface vertex in proximity to the metal artefact of contact of the electrode (Randazzo et al., 2016). The reviewer is right that this does not give a definitive anatomical verification of the location, but we are confident that the correct location of each contact on individual gyri can be determined with acceptable certainty. Nevertheless, we have added a brief paragraph to acknowledge the limitation:

“We acknowledge that electrode locations were determined indirectly, as our intraoperative imaging modality was two dimensional, however our two-dimensional to three-dimensional fusion technique previously was shown to localize ECoG recording locations with high functional-anatomic accuracy (Randazzo et al., 2016).”

8) Please add trial numbers for each participant.9) Given that narrowband γ activity has been associated with pathophysiology (i.e. involuntary movements), please add information about the relationship between the disease state/scores and the magnitude of the coupling. Is this event-related coupling modulated at all by disease or is it purely a movement-related effect?

From our data it is unfortunately not possible to tell if coupling would become weaker as the disease progresses or if it was stronger in those patients who were more prone to dyskinesia because the coupling strength varies strongly with the nature of the captured units. For example, within one patient, who contributed 5 different units, the coupling strength was highly variable between the different cells, partly because of a difference in spike numbers but also because of a genuine variability in coupling strength between cells. A correlation with disease, tremor or dyskinesia severity could only be meaningfully computed if a large enough number of cells would be available for each patient to compute an average coupling score for each patient before attempting a correlation.

Furthermore none of the patients were dyskinetic as they were off levodopa when they were recorded.

10) Figure 2C. Was the firing rate analyzed using the same variable windows described for coupling analysis? Or do they report instantaneous firing rate? Would results be different if alternative method was used?

We used the variable window procedure only to compute the ISI CV, percentage bursting and the ISI mode among the firing characteristics. For the firing rates we had to use a different procedure, as by design the variable window procedure would return the same firing rate for each window (that are scaled in size to contain the same number of spikes). To make it comparable, though, we computed the firing rates within a 0.3s window around each time point, because the average window in the other procedure would also span 0.3s.

We state that at the very beginning of the Materials and methods section on Firing characteristics:

“Firing rates were calculated within a 300ms window around each time point to get a time-evolving estimate.”

If we would compute instantaneous firing rates, the results would look similar (see the right column in Author response image 3 compared to Figure 2C from the paper to the left). We have also added the following sentence to the Results: “This was the case not only when computing average firing rates in 0.3s long sliding windows as shown in Figure 2C, matching the average window length used to calculate the coupling strength, but also when examining instantaneous firing rates.”

**Author response image 3. respfig3:** Comparison of analyses using 300ms average versus instanteous firing rate.

[Editors' note: further revisions were suggested prior to acceptance, as described below.]

Essential revisions:– Although improved, many aspects of the Materials and methods and Results are still very difficult to understand. It is often difficult to understand what data went into what analyses and what each analysis reflects. One option would be to include a schematic figure to illustrate this.

Thank you for raising this issue. To provide a better overview what each figure reflects and what answers can be drawn from them, we have now also added a new figure (Figure 7), which highlights the questions we covered in this study.

We have also removed some of the analyses that only considered a subset of the data (further detailed below).

Also, in the Results, it might be helpful to move some of the details to the Materials and methods and try to explain the analyses more conceptually – i.e. what was the analysis testing [i.e. what do the results mean] and what did you find? For instance, some of the detail in subsection “STN spiking probability is modulated relative to the cortical γ phase” seems more appropriate for methods.

We have removed the details from the Results and have ensured that the procedure is explained in detail in the Materials and methods. The section is now shorter and easier to read. We have also moved the right column of Figure 5 (old Figure 4) to the supplementary figures, because all other results based on this alternative set of ECoG sites are now also only shown in the supplementary figures.

– For the electrode selection method – please add a bit more detail. The authors state that the electrodes with strongest spike-to-γ coupling were used. Strongest compared to what? Strongest per subject? Strongest relative to group? It would be helpful to get a sense of what the other electrodes showed? Was there no coupling? Or did all electrodes show some coupling, but not as much. Perhaps at least an example of coupling for all electrodes in an example subject would help give a better sense of the data.

It was strongest spike-to-γ coupling per spike recording. We have now added in the Results section:

“The bipolar contact pairs with the strongest cortical γ phase coupling to STN spikes during contralateral gripping per spike recording were concentrated in lateral precentral gyrus.”

And in the Materials and methods:

For each spike recording, the ECoG contact with the highest PLV between 60-80 Hz and within -0.1 – 0.4s around movement onset during contralateral gripping was pre-selected for further analyses.

We have now also added an example recording as Figure 2—figure supplement 1 to show the distribution of coupling strengths during contra- and ipsilateral gripping. This figure also shows that there may have been more than one site showing significant coupling.

– For the comparison of phase offset between contra- vs ipsi- gripping, one concern is if the 180 degree difference could be due to the bipolar montage (i.e. signals were flipped for a trivial reason). How do the authors know that this is not the case? (Since they describe in detail the need to flip signals for the HFA analysis). We may be mis-understanding here…

Thank you for pointing out that this was unclear. The comparison between the preferred phase of spikes during contra- and ipsilateral gripping did not depend on the signal-flipping procedure because the polarity-standardization procedure was performed on the whole recording, hence the bipolar montage was the same for contra- and ipsilateral grip trials. Note that for this analysis, fixed phase offsets are irrelevant: When computing the relative difference between the preferred phase during contra- and ipsilateral gripping, a difference of, for example, one pi, would remain the same irrespective of the polarity. If the ECoG phase vector values would all be shifted by +pi (which is what happens when flipping the polarity), then the preferred phase during contralateral gripping would change for example from 1.5pi to 2.5pi and for ipsilateral gripping, for example from 0.5pi to 1.5 pi. Hence, the circular distance (1.5pi – 0.5pi = 1pi and 2.5pi-1.5pi = 1pi) would remain the same.

We have now added a more detailed explanation in the Materials and methods section:

Another key question was whether the timing of STN spikes relative to the cortical γ phase systematically differed between contra- and ipsilateral grip trials. Note that because the polarity of the γ signal was standardized based on the HFA from the whole recording, the polarity was the same for contra- and ipsilateral grip trials. Because here we directly compare the phase difference between contra- and ipsilateral gripping, a difference of 1pi, for example, would remain the same even if the polarity of the whole signal would be flipped, as may occur with bipolar derivations.

– Analysis in Figure 5C seems a bit biased. Since electrodes were selected based on strongest coupling in this time-range, is it surprising that the phase is most consistent there as well? Other time periods would be expected to have systematically less coupling in general, right? Perhaps because they have inconsistent phase? This should be corrected or at least, this potential confound should be discussed.

Figure 5C mainly served to double-check if the effect is indeed specific to movement, where we observed stronger coupling. We have now modified the Results section:

“P-values derived from a V-test assessing directionality towards an offset of 180° showed that, as expected, the significant phase difference was only present around movement onset (Figure 6C), where coupling was strongest.”

The preferred spike timing per se for individual recordings, which we compared in Figure 5C, is independent of the coupling strength, so the fact that the effect is present in this window does not merely stem from the electrode selection. However, as the reviewers pointed out, the fact that we don’t see it in other windows is also linked to the fact that there was no consistent coupling.

Now we also acknowledge in the Materials and methods:

“A V-test examines the phase differences for circular uniformity, similarly as a Rayleigh test, but is more powerful because a mean difference – in this case 180° – can be specified. Note that the average preferred phase, as compared here, is independent of the coupling strength used to select the ECoG sites. Detecting a significant effect would thus be unrelated to the ECoG selection procedure, however because coupling was strongest around movement onset, the effect would be expected to be temporally specific to windows around movement onset.”

– In the Materials and methods it is mentioned that the ECoG is filtered with a bandpass cutoff of 250 Hz, so how can the HFA include activity at >300 Hz? Further to this point, from the Materials and methods, it sounds as the filtering has been done "online". From the writing it is not clear if the entire filtering (notch filtering and also the band pass filtering) was done online or only the notch filtering was done online. If former is the case, doesn't this mean that the recorded data was filtered at 250 Hz and then stored by the recording system? Then how was it possible to analyze >300 Hz activity if it was above the filtering cut off? Please clarify this point. Also please add filter details for bandpass filter.

According to the Ripple Neuro support team, the online 250 Hz low-pass 4^th^ order Butterworth filter attenuates 250 Hz signals by 3 dB and signals at 400 Hz by 16 dB. So despite strong attenuation, the signal still contains information. However, we agree that it is preferable to perform the analyses of the high-frequency activity on unfiltered high-resolution data, which is why we have now retrieved the raw data sampled at 30 kHz and performed the analysis of 300 Hz high-pass filtered activity on this broadband data.

We have also added the filter details to the Materials and methods:

“ECoG signals were amplified, online notch filtered (at 60, 120, and 180 Hz, 2^nd^ order Butterworth filter), online bandpass filtered (0.3–250 Hz, 4^th^ order Butterworth filter) and digitized at 1,000 Hz using the Grapevine NIPS. In addition to the down-sampled data, a broadband version of the data was also recorded at 30 kHz (low-pass filtered with a 3rd order Butterworth anti-aliasing filter at 7500 Hz), which was used for analyses of 300 Hz high-pass filtered high-frequency activity.”

– The rationale for calculating HFA is not clear. The authors cite Logothetis, 2003 for the selection of the 300 Hz cutoff, however, it seems this paper used the 300 Hz cutoff for multi-unit activity, and here the authors are using ECoG. STN LFPS focus on higher ranges, but in ECoG, broadband γ as a surrogate for neural firing extend much lower (extending down to 70 Hz), and have generally not been reported >300 Hz. If the broadband signal really does extend this low, there would be overlap with the narrowband signal, which would complicate the coupling analysis. Finally, could the authors clarify how the use of the full wave reactivation compares to taking the analytic amplitude of the signal? It was not clear why rectification was used as the analytic amplitude approach would be more conventional and perhaps reflect more readily the author's goals (if we understand correctly)? In short, this portion of the paper is hard to understand. Please clarify or perhaps remove from the manuscript.

How the HFA evolved relative to the γ phase in individual patients looked similar when using the 30 kHz data compared to the 1000 Hz data. In Author response image 4 we show two example recordings, which also show that taking the amplitude of the analytic signal (3^rd^ column) or taking the >150 Hz high-pass filtered activity instead of 300 Hz (4^th^ column) would result in the same decision regarding the polarity. In fact, for all the 28 recordings, the polarity standardization results based on the 30 kHz data would not change if we would use the latter two, which we now also point out in the Materials and methods:

“A 150 Hz high-pass filter, or alternatively computing the amplitude of the analytic signal instead of full-wave rectification, would have provided the same results.”

Taking the amplitude of the analytic signal or full-wave rectifying the signal would thus both be valid options. Because low-pass filtering is performed as a next step, the choice would make barely any difference. We decided to stay with the latter because the convention in past studies was also to use full-wave rectification (Eckhorn et al., 1988; Pooresmaeili et al., 2010).

**Author response image 4. respfig4:** The first column shows two examples of the HFA based on the 1000 Hz data, which was used in the previous version of our article. The second column shows HFA based on the 30 kHz data, which is what we have used now for the revised version.

We have also retained 300 Hz as reported threshold, because the principle of using 300 Hz high-pass filtered activity as a proxy of background firing is the same irrespective of using microelectrodes arrays or ECoG strips. As the reviewers pointed out, using for example >70 Hz filtered activity as HFA would result in an overlap with the narrowband signal. It would be more difficult to detect nesting of HFA within the γ cycle if we would have considered such low activity as HFA, because power fluctuations would likely be dominated more strongly by the lower frequencies where power fluctuations are slower and would thus unlikely show clear co-fluctuations with the 60-80 Hz phase. Using the 300 Hz cut-off thus made sure that our HFA was far away from the 60-80 Hz band.

We would like to acknowledge that, as the reviewers state, broadband high-frequency activity as a surrogate for neural firing can extend down to 70 Hz (which we also acknowledged in the discussion). However, now we do not report the old Figure 6 anymore, where we had shown the average HFA as a proxy of firing co-fluctuating with the γ phase. Now we only use the HFA as a tool to standardize the polarity.

We have added this to the Materials and methods:

“Using a cutoff of 300 Hz made sure that we could detect fast fluctuations of activity. This would have been more difficult if a cutoff in the vicinity of 60-80 Hz were used because the signal would then likely be dominated by slower fluctuations in lower frequencies.”

Using the 30 kHz data instead of the 1000 Hz data, however, did result in a different polarity for 7 out of 28 recordings. See for example, the following case in Author response image 5:

**Author response image 5. respfig5:** Comparison of analysis using data sampled at 1000Hz versus 30kHz.

This resulted in slightly different p-values for the comparisons of the spiking probability across bins. We have thus modified the analysis to be computed across 4 phase bins instead of 5 phase bins (see updated Results and Materials and methods sections). The modulation across 5 bins still did reach significance (as shown in Figure 5—figure supplement 3), however, although the difference of probabilities in individual bins between contra and ipsilateral gripping was less pronounced than found previously.

The analyses of the steepness of the HFA increase and the CIs for the phase difference formerly reported in the old Figure 6 were not significant in the new analyses based on 30 kHz data. Hence all sections on this topic were removed from the revised manuscript. The only effect we have still included is that the HFA was significantly higher during contralateral vs. ipsilateral gripping in the beginning of the Results section.

– The Authors mention that they use notch filter to at 60 Hz to remove line noise. Please specify the notch filter characteristics. This is important given that the 60-80 Hz frequency range is used to analyze γ activity which partly includes the line noise frequency range.

We have now added the information that it was 2^nd^ order Butterworth filter in brackets. The filter resulted in a very focussed attenuation of the line noise, as you can see in the examples of two power spectra in Author response image 6:

**Author response image 6. respfig6:** Line-noise attenuation.

– Some of the interpretations in subsection “Cortical high-frequency activity increases faster in sites that show strongest γ coupling during contralateral gripping” seem very definitive when maybe they are really more hypotheses. The authors may consider toning town the language a bit.

Thank you for pointing this out. This section has now been removed.

– For our original point 9 – The author's explanation makes sense, but the variability of relationships between unit to ECoG coupling within a person should be stated somewhere in the manuscript. The authors may also want to mention that this is why across subject's correlations related to clinical severity cannot be examined.

We have now added to the Discussion:

“Note that because coupling strength varies strongly with the nature of the captured units, and thus different units recorded from one patient can result in variable coupling values, we did not perform correlations between coupling strength and disease severity. To meaningfully compute such correlations, the number of cells sampled from each patient would need to be large enough to compute a reliable mean coupling score.”

– Please put the table with patient characteristics and experimental information back in main manuscript (not supplements).

We have now moved it into the main manuscript.

– It would be preferable to include the task schematic and a bit more description in the main paper, not just the supplements.

We have added it as Figure 1 to the main paper and provided more information in the figure caption.

– It would be helpful to provide a bit more context about the Cheyne paper and why it was selected for this comparison. Also, although it is somewhat obvious given the paper was published 11 years ago, it might be helpful to clearly state that this was an independent paper, in different participants. Also, related to this, the authors refer to "detected in MEG studies", but then cite just one paper. Perhaps change to "detected in a MEG study".

We have changed it to “detected in a previous MEG study” and have also changed the description in the figure caption to “A magnetoencephalography study in healthy participants showed independently from our study that γ oscillations at the onset of finger movements are focal to lateral motor cortex (Cheyne et al. 2008).”

We now also elaborate further in the Results section:

“First, we assessed if movement-related STN spike-to-cortical γ coupling was specific to a region of primary motor cortex that has also been demonstrated to show the strongest movement-related γ power increase in MEG recordings (Figure 2B, adapted from Cheyne et al., 2008). This MEG study has shown a contralateral γ increase at highly specific spatial locations for individual limb movements, which was highly consistent over repeated measurements and sessions (Cheyne et al., 2008; Cheyne, 2013).”